# GUARANTEED USER FAIRNESS IN RECOMMENDATION

## ABSTRACT

Although recommender systems (RS) have been well-developed for various fields of applications, they suffer from the crisis of platform credibility with respect to RS confidence and fairness, which may drive users away from the platform and result in the failure of the platform's long-term success. In recent years, a few works have tried to solve either the model confidence or fairness issue, while there is no statistical guarantee for these methods. Therefore, there is an urgent need to solve both issues with a unifying framework with statistical guarantees. In this paper, we propose a novel and reliable framework called Guaranteed User Fairness in Recommendation (GUFR) to dynamically generate prediction sets for users across various groups, which are guaranteed 1) to include the ground-truth items with user-predefined high confidence/probability (e.g., 90%); 2) to ensure user fairness across different groups; 3) to have the minimum average set size. We further design an efficient algorithm named Guaranteed User Fairness Algorithm (GUFA) to optimize the proposed method, and upper bounds of the risk and fairness metric are derived to help speed up the optimization process. Moreover, we provide rigorous theoretical analysis concerning risk and fairness control and the minimum set size. Extensive experiments also validate the effectiveness of the proposed framework, which aligns with our theoretical analysis. The code is publicly available at https://anonymous.4open.science/r/GUFR-76EC.

## 1 INTRODUCTION

Recommender Systems (RS) (Aggarwal, 2016; Fan et al., 2022; Sharma et al., 2024)are a type of information filtering system designed to provide suggestions to users based on their preferences. Decades of effort have been devoted to improving the accuracy of these recommendation models, However, less attention has been paid to model confidence, affecting users' trust in the platform's credibility. In recent years, few recommendation approachesNaghiaei et al. (2022); KWEON et al. (2024) are developed for model confidence. However, these methods are heuristic modeling without statistical guarantee. Meanwhile, fairness is another critical issue that could lead to poor user experience on the platform's reliability. Some fairness-based recommendation models Li et al. (2023) have been developed in recent years Han et al. (2023; 2024b). While these papers effectively alleviate fairness issues in recommendation systems, they are typically empirically validated without statistical guarantees for both the performance and fairness.

As a result, we are motivated to develop a complete and statistically guaranteed recommendation framework that considers both model confidence and fairness issues as a whole in this paper. Our overall goal is to construct set predictors that can generate the minimum prediction set for each user while being guaranteed to provide model confidence and ensure user fairness among different groups. Thus, the objectives of our framework are threefold: (1) to construct prediction sets that cover the true item with a high user pre-defined probability, say 90% (i.e., confidence level); (2) to guarantee user fairness across different groups; and (3) to guarantee the minimum average set size while ensuring (1) and (2).

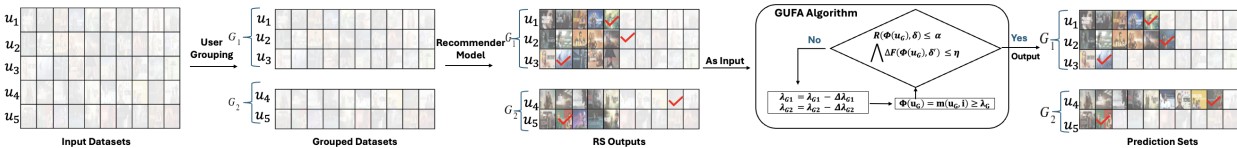

**Figure 1:** The proposed GUFR Framework. Red check marks indicate the true relevant items.

Inspired by Risk-Controlling Prediction Sets (RCPS) Bates et al. (2021b) - a powerful statistical tool, we propose a reliable and fair framework called Guaranteed User Fairness in Recommendation (GUFR) to achieve the above-mentioned objectives. Even though we can directly apply the framework of RCPS to define the coverage guarantee via the risk of not including the ground-truth item to achieve the first objective. However, it is far from enough to achieve the rest objectives. There are several challenges remaining to be solved: 1) How to define user fairness among different groups in a statistical way? 2) How to produce prediction sets with minimum size? 3) How to improve the efficiency

of constructing prediction set when the search range is so large? 4) How to theoretically guarantee the constructed prediction sets to meet the risk control and fairness definition as well as the minimum set size? To address these gaps, we first define an estimator called fairness metric, which is required to meet the fairness-controlling prediction sets (FCPS) defined in a similar way as that of the risk control. We then build our objective function by minimizing the average prediction set while making it meet both RCPS and FCPS constraints for all users across different groups. Subsequently, we derive the upper bounds for both the risk and fairness to accelerate the optimization process for prediction set construction. Lastly, we provide theoretical analysis to prove the effectiveness of set predictors with respect to RCPS and FCPS, and the minimum set size. The proposed framework is depicted in Fig: 1.

Our contributions are summarized as follows:

- Firstly, we formulate the recommendation problem from statistically guaranteed perspectives in terms of the risk control guarantee and fairness control guarantee, and propose a reliable and fair recommendation framework, i.e., Guaranteed User Fairness in Recommendation (GUFR), which is able to construct minimum prediction set while ensuring the risk control and fairness guarantee for all users in different groups.

- Secondly, we design an efficient optimization algorithm, i.e., Greedy User Fairness Algorithm (GUFA) to optimize the objective function of GUFR. To accelerate the optimization process, we derive the upper bounds for both the defined expected risk and fairness metric via concentration inequalities in Theorem 1 and Theorem 2 and then make them approach their respective thresholds in a greedy way.

- Thirdly, we establish rigorous theoretical guarantees for the proposed framework GUFR. We prove that the constructed prediction set can achieve risk control and fairness guarantees in Theorem 3 while achieving minimal set sizes in Theorem 4, which theoretically verifies the effectiveness of GUFR.

- Finally, we conduct comprehensive experiments on top of five commonly used recommendation models and various datasets across multiple domains, the results of which demonstrate the effectiveness of the proposed GUFR empirically, which aligns with our theoretical analysis.

## 2 RELATED WORKS

### 2.1 RECOMMENDATION

Recommender systems (Ko et al., 2022; Lu et al., 2015), which are used to help users make decisions via personalized content or product recommendations, have been thoroughly studied over the past decades by developing recommendation models for different fields of application, such as e-commerce (Schafer et al., 1999), media streaming (Chang et al., 2017), social networks (He et al., 2024) etc. To ensure the long-term success of digital platforms, credibility and fairness of recommendation are two crucial factors today that are urgently needed to ensure the satisfaction of customers. Traditional recommendation models primarily focused on accuracy (Adomavicius & Tuzhilin, 2005; Ricci et al., 2010), but there is an increasing recognition that model confidence—quantifying the reliability of the recommendations—is equally important. Researchers try to develop recommendation approaches that provide the model confidence. For example, Kweon et.al. (2024) enhances model confidence by dynamically adjusting recommendation strategies based on performance metrics, ensuring the model adapts to user interactions and maintains high predictive accuracy. Naghiaei et al. (2022) proposes a confidence-aware optimization-based re-ranking algorithm that accounts for calibration confidence based on user profile size. However, these methods are heuristic modeling without statistical guarantee. Meanwhile, some fairness-based recommendation models have been developed in recent years, which usually focus on a particular fairness issue in specific fields of application. Fairness in recommendation systems can be viewed from diverse perspectives Li et al. (2023). One such perspective is Individual fairness and Group fairness. Individual fairness requires that similar individuals receive comparable treatment. However, defining this similarity is challenging due to disagreements over task-specific similarity metrics Dwork et al. (2011). Group fairness, on the other hand, ensures that protected groups receive treatment comparable to that of advantaged groups or the general population Pedreschi et al. (2009), thus ensuring equitable treatment across predefined groups. Group fairness can be further classified from the user side or item/platform side. Focusing on User-Side group Fairness, it can be defined based on sensitive features like age, gender, race, etc. Yao & Huang (2017) utilized gender to distinguish between advantaged and disadvantaged user groups and measured prediction discrepancies. Another approach utilizes differentiating groups based on user interactions as defined by Li et al. (2021a) and Abdollahpouri et al. (2019). They defined unfairness between users based on their interaction levels or popular item interactions, labeling the top 5% as active or advantaged. To ensure fairness, existing works apply several techniques, most commonly in regularization and constrained optimization Li et al. (2021a); Islam et al. (2021), where fairness is given as a regularization parameter. Furthermore, Han et al. (2024a) mitigate unfairness by sharing training samples within and across user groups, effectively addressing data sparsity. Another work by Han et al. (2024b) proposes a hypergraph-based framework that models high-order correlations among users to improve training for disadvantaged users. Some other approaches use Reinforcement Learning by formulating the problem as a Constrained Markov Decision Process Ge et al. (2021;

2022). To evaluate the fairness, Yao & Huang (2017) introduced four group metrics to evaluate collaborative filtering recommender models. Fu et al. (2020) employed the Group Recommendation Unfairness (GRU) metric to assess disparities across these user groups based on performance metrics. They depicted it balances fairness with utility—a dual improvement noted by Rahmani et al. (2022), under certain conditions. However, to the best of our knowledge, none of these approaches ensured a statistical guarantee of the fairness problem.

## 2.2 RISK-CONTROLLING PREDICTION SETS

We develop uncertainty quantification for the model confidence and fairness based on Risk-Controlling Prediction Sets (RCPS) Bates et al. (2021b). RCPS is a general framework, not a specific algorithm, for producing predictive sets that satisfy the risk control in Definition 2.1. Different contexts require different designs of risk or other estimators to achieve the best performance. For example, in the context of medical diagnosis, if the set $S(X)$ represents plausible diagnoses based on patient features $X$ and $R(S)$ can be defined as the expected risk of the loss from missing true diagnoses, then RCPS ensures this risk to remain below $\alpha$ with confidence $1 - \delta$. This enables doctors to automatically screen for many diseases (e.g., via a blood sample) and refer the patient to relevant specialists. We will apply the framework of RCPS to the designed risk and fairness in the context of recommendation.

**Definition 2.1** (Risk-controlling prediction sets (RCPS) Bates et al. (2021b))**.** *Let $\mathcal{S}$ be a random function taking values in the space of functions $\mathcal{X} \to \mathcal{Y}'$ (e.g., a functional estimator trained on data). We say that $\mathcal{S}$ is a $(\alpha, \delta) - RCPS$ if, with probability at least $1 - \delta$, we have $\mathcal{R}(\mathcal{S}) \leq \alpha$.*

## 3 THE PROPOSED FRAMEWORK

In this section, we formulate the objective functions that our framework, i.e., Guaranteed User Fairness in Recommendation (GUFR), aims to achieve. Firstly, we introduce the notations used in the paper. Consider $n$ items, denoted as $\boldsymbol{i} = [i]_{j=1}^n$, where each item $i_j$ is an element of the item space $\mathcal{I}$. Similarly, we have $m$ users, represented by $\boldsymbol{u} = [u]_{k=1}^m$, where each user $u_k$ belongs to the user space $\mathcal{U}$. For brevity, we use $u$ and $i$ for user and item, respectively. The group information $G$ of each user $u$ is known, and following (Li et al., 2021b), we partition users into two groups, $G_1$ and $G_2$, such that $G_1 \cap G_2 = \emptyset$ and $G_1 \cup G_2 = \mathcal{U}$ to ensure exclusivity. Here, $G_1$ and $G_2$ represent the advantaged and disadvantaged groups, respectively.

The recommendation is conducted via the relevance model $m : \mathcal{U} \times \mathcal{I} \to [0, 1]$, which maps a user $u$ and an item $i$ to an estimate score $m(u, i)$, and items with the highest scores are usually the most relevant recommendations. However, there is no theoretical guarantee to ensure the confidence of the model's output, and so the reliability of the recommended items remains uncertain. In the following, we will follow the framework of risk-controlling prediction sets (RCPS) Bates et al. (2021a) to solve this gap. We define our set predictor to be $\phi : u \to \boldsymbol{i}'$, where $\boldsymbol{i}' \subseteq \mathcal{I}$ is a set-valued output guided by parameter $\lambda$. This lambda takes values in a closed set $\Lambda \subset \mathbb{R}$ such that $\phi(.)$ is nested i.e.,

$$\lambda_1 < \lambda_2 \implies \phi_{\lambda_2}(u) \subset \phi_{\lambda_1}(u).$$

We further define the loss function between the relevant item $i_{\text{true}}$ of user $u$ and the prediction set $\phi_\lambda(u)$ to be 0-1 loss as follows:

$$L(i_{\text{true}}, \phi_\lambda(u)) = \begin{cases} 1 & \text{if } i_{\text{true}} \notin \phi_\lambda \\ 0 & \text{if } i_{\text{true}} \in \phi_\lambda. \end{cases} \quad (1)$$

It is worth noting that $L(i_{\text{true}}, \phi_\lambda(u))$ also meets the monotonicity (larger set leads to smaller loss), namely,

$$\phi_{\lambda_1}(u) \subset \phi_{\lambda_2}(u) \implies L(i_{\text{true}}, \phi_{\lambda_1}(u)) \geq L(i_{\text{true}}, \phi_{\lambda_2}(u)). \quad (2)$$

Based on the above loss function, we define the expected risk of not including a ground-truth item in the prediction set for all users as follows:

$$R(\lambda_G) = E(L(i_{\text{true}}, \phi_{\lambda_G}(u))). \quad (3)$$

Subsequently, we require the defined risk to meet the risk-controlling prediction sets (RCPS), which ensures the probability of the risk lower than a user-specified threshold $\alpha$ is no less than the user-defined confidence level $1 - \delta$, namely, the reliability of recommendation. The detailed formulation can be found as follows:

$$\Pr(R(\lambda_G) \leq \alpha)) \geq 1 - \delta. \quad (4)$$

Meanwhile, fairness of users in the advantaged groups and the disadvantaged groups is another tough issue that needs to be tackled. Thus, we define a fairness metric $\Delta F(\cdot)$ via the difference between the normally used recommendation metric (such as hit rate (HR) and NDCG) of the advantaged group $G_1$ and the disadvantaged group $G_2$, to evaluate user fairness as follows:

$$\Delta F(\lambda_{G_1}, \lambda_{G_2}) := \left| \frac{1}{|G_1|} \sum_{u \in G_1} M(\phi_{\lambda_{G_1}}(u)) - \frac{1}{|G_2|} \sum_{u \in G_2} M(\phi_{\lambda_{G_2}}(u)) \right|. \quad (5)$$

Here, $M(\cdot)$ denotes a generalized function representing a recommendation metric (such as HR or NDCG) that measures the performance of the recommendation set $\phi_{\lambda_G}(u)$ for any user $u$.

For example, when we use hit rate (HR) or NDCG as the recommendation metric, we can express them as:

$$\mathrm{HR}(G_i) = \frac{1}{|G_i|} \sum_{u \in G_i} \mathbb{I}(\text{relevant item in } \phi_{\lambda_{G_i}}(u)),$$

$$\mathrm{NDCG}(G_i) = \frac{1}{|G_i|} \sum_{u \in G_i} \mathrm{NDCG}.(\phi_{\lambda_{G_i}}(u)),$$

Thus, the fairness metrics can be expressed as:

$$\Delta_{\mathrm{HR}} = |\mathrm{HR}(G_1) - \mathrm{HR}(G_2)|, \quad \Delta_{\mathrm{NDCG}} = |\mathrm{NDCG}(G_1) - \mathrm{NDCG}(G_2)|.$$

This design makes our proposed framework more flexible by accommodating different types of RS metrics.

Similarly, we require the defined fairness metric to meet the fairness-controlling prediction sets (FCPS), that is, the probability of the fairness metric lower than a user-specified threshold $\eta$ is no less than user-pre-defined confidence level $1 - \hat{\delta}$, namely, the reliability of fairness. The detailed formulation can be expressed as follows:

$$\Pr(\Delta F(\lambda_{G_1}, \lambda_{G_2}) \leq \eta) \geq 1 - \hat{\delta}. \tag{6}$$

Moreover, we hope the constructed prediction sets to be as small as possible while they meet the risk-controlling guarantee as well as the fairness-controlling guarantee. This is because a smaller set size will contribute to less uncertainty of recommendation and improve the usablity and effectiveness of RS. Therefore, our goal is to find the optimal $(\lambda_{G_1}, \lambda_{G_2})$ that minimizes the average size of the recommendation sets, satisfying the risk (coverage) and fairness guarantees for all users in groups $G_1$ and $G_2$. The objective function can be formulated as follows:

$$
\begin{aligned}
\arg\min_{(\lambda_{G_1}, \lambda_{G_2})} &\sum_{G \in \{G_1, G_2\}} \frac{1}{|G|} \sum_{u \in G} |\phi_{\lambda_G}(u)| \\
\text{s.t.} \quad &\Pr(R(\lambda_G) \leq \alpha)) \geq 1 - \delta \text{ for all } G \in \{G_1, G_2\}, \\
&\Pr(\Delta F(\lambda_{G_1}, \lambda_{G_2}) \leq \eta) \geq 1 - \hat{\delta}.
\end{aligned}
\tag{7}
$$

Here, $\alpha$ and $\eta$ are the user pre-specified risk and fairness thresholds, say 10%; $1 - \delta$ and $1 - \hat{\delta}$ are the user pre-defined confidence level for the risk and fairness, say 90%.

## 4 THE OPTIMIZATION ALGORITHM

To optimize the objective function in eq. (7), we need to ensure the risk and fairness metric in the constraints are below decision-makers' pre-defined value $\alpha$ and $\eta$ respectively, and finally obtain the optimal prediction set with minimum size. It is not efficient to directly apply the greedy algorithm as the range of risk and fairness values that approach the threshold $\alpha$ and $\eta$ by adjusting the $(\lambda_{G_1}, \lambda_{G_2})$ is very large. If we can derive the upper bounds of both risk and fairness metric and take values at their corresponding upper bounds $R_G^+(\lambda_G, \delta)$ and $\Delta F^+(\lambda_{G_1}, \lambda_{G_2}, \hat{\delta})$ respectively, then it will become more efficient to approach the threshold $\alpha$ and $\eta$ by adjusting the $(\lambda_{G_1}, \lambda_{G_2})$. Following the upper bound strategy to accelerate the optimization procedures in Bates et al. (2021b), we have the optimized risk constraint as follows:

$$\Pr(R(\lambda_G) \leq R^+(\lambda_G, \delta)) \geq 1 - \delta \quad \text{and} \quad R_G^+(\lambda_G, \delta) \leq \alpha \quad \text{for all} \quad G \in \{G_1, G_2\}. \tag{8}$$

Similarly, the optimized fairness metric constraint can be reformulated as follows:

$$\Pr(\Delta F(\lambda_{G_1}, \lambda_{G_2}) \leq \Delta F^+(\lambda_{G_1}, \lambda_{G_2}, \hat{\delta})) \geq 1 - \hat{\delta} \quad \text{and} \quad \Delta F^+(\lambda_{G_1}, \lambda_{G_2}, \hat{\delta}) \leq \eta. \tag{9}$$

Consequently, we can choose $\hat{\lambda}$ as the largest value of $\lambda$ such that the entire confidence region to the left of $\lambda$ falls below the target risk level $\alpha$ and $\eta$, and the set size will achieve the minimum value. The optimized objective function can be formulated as follows:

$$(\hat{\lambda}_{G_1}, \hat{\lambda}_{G_2}) = \sup\left\{\lambda_{G_1}, \lambda_{G_2} \in [0, 1] : R^+(\lambda_G, \delta) \leq \alpha, \quad \Delta F^+(\lambda_{G_1}, \lambda_{G_2}, \hat{\delta}) \leq \eta\right\}. \tag{10}$$

---

**Algorithm 1** Greedy User Fairness Algorithm (GUFA)

---
1: **Initialization:**
2: Initialize control parameters for the two groups $\lambda_{G_1}, \lambda_{G_2}$
3: Initialize user pre-specified parameters $\alpha, \eta, \delta, \hat{\delta}$
4: **Define** Loss and Risk as in eqs. (1) and (4)
5: **Define** Fairness metric as in eq. (5)
6: **Adjustment Loop:**
7: **for** users in each group $G \in \{G_1, G_2\}$ **do**
8:      Calculate $R_G^+(\lambda_G, \delta)$ such that $\Pr(R(\lambda_G) \leq R_G^+(\lambda_G, \delta)) \geq 1 - \delta$
9:      Compute $\Delta F(\lambda_{G_1}, \lambda_{G_2})$ and calculate $\Delta F^+(\lambda_{G_1}, \lambda_{G_2}, \hat{\delta})$ such that $\Pr(\Delta F(\lambda_{G_1}, \lambda_{G_2}) \leq$
     $\Delta F^+(\lambda_{G_1}, \lambda_{G_2}, \hat{\delta})) \geq 1 - \delta$
10:      **if** $R_G^+(\lambda_G, \delta) > \alpha$ OR $\Delta F^+(\lambda_{G_1}, \lambda_{G_2}, \hat{\delta}) > \eta$ **then**
11:         Update $\lambda_{G_1} \leftarrow \lambda_{G_1} - \Delta_1, \lambda_{G_2} \leftarrow \lambda_{G_2} - \Delta_2$
12:      **end if**
13: **end for**
14: $\hat{\lambda}_{G_1}, \hat{\lambda}_{G_2} \leftarrow \lambda_{G_1}, \lambda_{G_2}$                      ▷ Get the optimal $\lambda_{G_1}, \lambda_{G_2}$
15: **Construct Prediction Sets:**
16: **for** each user $u$ in group $G$ **do**
17:      $\phi_{\hat{\lambda}_G}(u) \leftarrow \{i \mid m(u, i) \geq \hat{\lambda}_G\}$
18: **end for**
19: **Output:** the optimal solution $\hat{\lambda}_{G_1}, \hat{\lambda}_{G_2}$ and prediction sets $\phi_{\hat{\lambda}_{G_1}}(u)$ and $\phi_{\hat{\lambda}_{G_2}}(u)$ for all users in different groups.

---

To optimize the above objective function and output the optimal solution for $(\hat{\lambda}_{G_1}, \hat{\lambda}_{G_2})$ that dominate the validity of set predictor, we design a novel greedy-strategy-based algorithm called Greedy User Fairness Algorithm. The complete procedures of the optimization algorithm are summarized in Algorithm 1.

However, it still remains unknown that what the upper bounds of risk and fairness metric look like. In the following part, we will derive the upper bounds in theorem 1 and theorem 2 respectively.

**Theorem 1** ( Upper Bound for Risk). *Assume the loss function $L(i_{true}, \phi_{\lambda_G}(u))$ follows a Bernoulli distribution, then the upper bound for the risk $R(\lambda_G)$ can be found as follows:*

$$R^+(\lambda_G, \delta) = \sup \left\{ \hat{R}(\lambda_G) : BinomCDF(n\hat{R}(\lambda_G), n, \alpha) \leq \delta \right\}, \tag{11}$$

*where $n$ is the number of samples; $G \in \{G_1, G_2\}$; $\hat{R}(\lambda_G)$ denotes the empirical risk of $R(\lambda_G)$, which can be calculated as follows:*

$$R(\lambda_G) = \frac{1}{|G|} \sum_{u \in G} L(i_{true}, \phi_{\lambda_G}(u)). \tag{12}$$

*Here, $|G|$ denotes the number of users in group $G$.*

*Proof.* Proof can be found in Appendix A.2.1. □

**Theorem 2** (Upper Bound for Fairness Metric). *The upper bound for fairness metric $\Delta F(\lambda_{G_1}, \lambda_{G_2})$ can be derived by applying Bernstein inequality Maurer & Pontil (2009) as follows:*

$$\Delta F^+(\lambda_{G_1}, \lambda_{G_2}, \hat{\delta}) = \Delta F(\lambda_{G_1}, \lambda_{G_2}) + \sqrt{\frac{2\sigma_F^2 \log\left(\frac{2}{\hat{\delta}}\right) + \frac{2}{3} \log\left(\frac{2}{\hat{\delta}}\right)}{n_1 + n_2}}, \tag{13}$$

*where $n_1$ and $n_2$ denote the number of samples for group $G_1$ and $G_2$; $\sigma_F^2$ denotes the variance associated with the fairness metric $\Delta_{HR}$ or $\Delta_{NDCG}$. The detailed formulation of the variance can be referred to in Appendix A.1.*

*Proof.* Proof can be found in Appendix A.2.2. □

**Recommendation** After obtaining the optimal $(\hat{\lambda}_{G_1}, \hat{\lambda}_{G_2})$ from Algorithm 1, we can recommend items for new customers. For example, when a new user $u_t$ comes, we first decide the group $G$ that they belong to, and then utilize the corresponding $\hat{\lambda}_G$ to calculate their prediction set via step 17.

## 5 THEORETICAL ANALYSIS

In this section, we provide theoretical analysis on the risk and fairness control guarantee in Theorem 3, as well as the minimum set size guarantee in Theorem 4.

**Theorem 3** (Risk and Fairness Control Guarantee). *For all group $G \in \{G_1, G_2\}$ and $\delta \in (0, 1)$, with probability of at least $1 - \delta$ for risk threshold $\alpha$, and with probability of at least $1 - \hat{\delta}$ for fairness threshold $\eta$, we have:*

$$\Pr(R(\hat{\lambda}_G) \leq \alpha) \geq 1 - \delta \wedge \Pr(\Delta F(\hat{\lambda}_{G_1}, \hat{\lambda}_{G_2}) \leq \eta) \geq 1 - \hat{\delta}. \tag{14}$$

*Proof.* Proof can be found in Appendix A.2.3. □

**Remark.** In Theorem 3, we prove that the optimal $\hat{\lambda}_{G_1}, \hat{\lambda}_{G_2}$ obtained from algorithm 1 are indeed able to control the expected risk to below the decision makers' defined values of $\alpha$ with confidence $1 - \delta$, and control the fairness metric $\Delta F$ to below the decision makers' defined values of $\eta$ with confidence $1 - \hat{\delta}$. This theoretically validate the recommendation reliability and fairness of the proposed GUFR framework.

**Theorem 4** (Minimum Set Size Guarantee). *Let $(\phi_{\lambda^*_{G_1}}, \phi_{\lambda^*_{G_2}})$ be any set predictor and let $(\phi_{\hat{\lambda}_{G_1}}, \phi_{\hat{\lambda}_{G_2}})$ be the optimal predictor obtained from Algorithm 1 such that $R(\lambda^*_G) \leq R(\hat{\lambda}_G)$ and $\Delta F(\lambda^*_{G_1}, \lambda^*_{G_2}) \leq \Delta F(\hat{\lambda}_{G_1}, \hat{\lambda}_{G_2})$. Then for each $G \in \{G_1, G_2\}$, we have:*

$$E\left[|\phi_{\hat{\lambda}_G}(u)|\right] \leq E\left[|\phi_{\lambda^*_G}(u)|\right]. \tag{15}$$

*where $|\phi_{\hat{\lambda}_G}(u)|$ denotes the predicted set size for any user $u$ in group $G$.*

*Proof.* Proof can be found in Appendix A.2.4. □

**Remark.** In Theorem 4, we prove that set predictor learned by our algorithm can output the minimal prediction set size for any user $u$ in group $G$, which theoretically validate the effectiveness of the proposed GUFR framework.

To sum up, set predictors constructed by Algorithm 1 can modify any black-box recommendation models to output prediction sets for new customers that are strictly guaranteed to satisfy the risk control as defined in equation 4 and the fairness control defined in equation 6 while ensuring the minimum prediction sets in equation 7.

## 6 EXPERIMENTS

In this section, we conduct experiments to validate the effectiveness of the proposed framework (GUFR). We design experiments to 1) validate whether the framework can provide the desired coverage guarantee in terms of risk, better performance in terms of average set size, and improved fairness in terms of Hit Rate Difference (Hit Rate Diff.) and NDCG Difference (NDCG Diff.) across various datasets with sensitive attributes; 2) analyze how the parameters ( i.e. $\alpha, \delta, \eta$ and $\hat{\delta}$) influence the performance; 3) analyze the time-efficiency of GUFR compared to other fairness baselines.

### 6.1 DATASETS AND BASE MODELS

We conduct experiments on four datasets with specific sensitive user attributes: (1) AmazonOffice dataset (eCommerce) (McAuley et al., 2015) grouped by item interactions; (2) Last.fm dataset (music streaming) (Cantador et al., 2011) grouped by region (developed and other countries; (3) MovieLens dataset (movie ratings) (Harper & Konstan, 2015) grouped by gender; and (4) Book-Crossing dataset (book ratings) (Ziegler et al., 2005) grouped by age. We implement the proposed framework on five base recommendation models: DeepFM (Guo et al., 2017), GMF (Koren et al., 2009), MLP (Zhang et al., 2019), NeuMF (He et al., 2017), and LightGCN (He et al., 2020). Additionally, we compare our framework GUFR with four fairness baselines: 1) NFCF (Islam et al., 2021) 2)MFCF (Islam et al., 2021) 3) GMF-UFR (Li et al., 2021a) 4) NCF-UFR (Li et al., 2021a). Details of all the datasets, base models, and fairness baselines can be found in A.3.

### 6.2 IMPLEMENTATION DETAILS

All base recommender models are trained for 20 epochs with a batch size 256, a learning rate of 0.001, the Adam optimizer, and Binary Cross Entropy Loss (BCELoss). For the NFCF and MFCF models, we modified the original code to generalize grouping logic for diverse criteria (e.g., interaction count, age, gender, and geography) and adapted the debiasing process to compute bias directions dynamically for various groups. For GMF-UFR and NeuMF-UFR,

to ensure consistency, we reused the score files generated by our base models. To enhance the reproducibility of the results, we utilized MIP Santos & Toffolo (2020), a free light-weight python library for modeling and optimization instead of Gurobi Gurobi Optimization, LLC (2024) optimization solver, a commercially licensed software. To ensure fair and sound comparisons with the base models and fairness baselines, instead of using arbitrary top-k predictions, we utilized the average optimal prediction set size returned by the GUFR framework on top of the given base recommendation model.

## 6.3 EXPERIMENTAL RESULTS

### 6.3.1 EXPERIMENTAL RESULTS W.R.T PERFORMANCE AND FAIRNESS

We compare the performance and fairness of the GUFR framework with five base recommendation models and four fairness baselines. We set the predefined risk threshold $\alpha = 0.2$, fairness threshold $\eta = 0.20$ via manual validation. The error rates $\delta = 0.1$ and $\hat{\delta} = 0.1$ are representatively set following Bates et al. (2021b). The coverage guarantee is measured in terms of risk; performance is measured using average set size, and fairness is compared using disparity in these metrics between user groups (Difference in Hit Rate and Difference in NDCG). The results for the AmazonOffice dataset (grouped by interactions), MovieLens dataset (grouped by gender), Last.fM dataset (grouped by region) and Book-Crossing dataset (grouped by age) are provided in Tables 1 to 4 respectively.

| Method | Group | Risk ↓ | Average Set Size ↓ | Hit Rate | NDCG | Hit Rate Diff ↓ | NDCG Diff ↓ |
|---|---|---|---|---|---|---|---|
| DeepFM | 1 | 0.121 | | 0.879 | 0.418 | 0.155 | 0.17 |
| | 2 | 0.277 † | 34 | 0.723 | 0.248 | | |
| DeepFM + GUFR | 1 | 0.192 | | 0.808 | 0.401 | 0.081 | **0.103** |
| | 2 | 0.111 | | 0.889 | 0.298 | | |
| GMF | 1 | 0.149 | | 0.851 | 0.439 | 0.212 † | 0.225 † |
| | 2 | 0.361 † | 30 | 0.639 | 0.214 | | |
| GMF + GUFR | 1 | 0.197 | | 0.803 | 0.428 | 0.08 | 0.168 |
| | 2 | 0.117 | | 0.883 | 0.26 | | |
| LightGCN | 1 | 0.077 | | 0.923 | 0.477 | 0.126 | 0.238 † |
| | 2 | 0.203 † | 39 | 0.797 | 0.239 | | |
| LightGCN + GUFR | 1 | 0.087 | | 0.913 | 0.474 | 0.087 | 0.198 |
| | 2 | 0 | | 1 | 0.276 | | |
| MLP | 1 | 0.162 | | 0.838 | 0.409 | 0.219† | 0.19 |
| | 2 | 0.38 † | **26** | 0.62 | 0.219 | | |
| MLP + GUFR | 1 | 0.197 | | 0.803 | 0.397 | **0.013** | 0.14 |
| | 2 | 0.184 | | 0.816 | 0.257 | | |
| NeuMF | 1 | 0.155 | | 0.845 | 0.414 | 0.225 † | 0.185 |
| | 2 | 0.379 † | 28 | 0.621 | 0.229 | | |
| NeuMF + GUFR | 1 | 0.182 | | 0.818 | 0.406 | 0.017 | 0.143 |
| | 2 | 0.199 | | 0.801 | 0.263 | | |
| Other Fairness Baselines | | | | | | | |
| NFCF | 1 | 0.196 | | 0.804 | 0.391 | 0.115 | 0.134 |
| | 2 | 0.261† | 28 | 0.689 | 0.257 | | |
| MFCF | 1 | 0.175 | | 0.825 | 0.402 | 0.128 | 0.154 |
| | 2 | 0.303 † | 30 | 0.697 | 0.248 | | |
| NeuMF-UFR | 1 | 0.193 | | 0.807 | 0.396 | 0.153 | 0.127 |
| | 2 | 0.346 † | 28 | 0.654 | 0.269 | | |
| GMF-UFR | 1 | 0.205 † | | 0.795 | 0.395 | 0.133 | 0.157 |
| | 2 | 0.368 † | 30 | 0.662 | 0.238 | | |

**Table 1:** Performances and fairness comparisons with base models and fairness baselines on the **AmazonOffice Dataset** grouped by the **Interactions** in terms of risk, average set size, and Hit Rate Diff/NDCG Diff, respectively. Bold indicates the best result, underline indicates the second best and † marks threshold exceeded cases.

| Method | Group | Risk ↓ | Average Set Size ↓ | Hit Rate | NDCG | Hit Rate Diff ↓ | NDCG Diff ↓ |
|---|---|---|---|---|---|---|---|
| DeepFM | 1 | 0.2 | | 0.8 | 0.503 | 0.017 | 0.022 |
| | 2 | 0.183 | 9 | 0.817 | 0.525 | | |
| DeepFM + GUFR | 1 | 0.188 | | 0.812 | 0.504 | 0.002 | 0.018 |
| | 2 | 0.187 | | 0.813 | 0.522 | | |
| GMF | 1 | 0.147 | | 0.853 | 0.538 | 0.051 | 0.019 |
| | 2 | 0.198 | 9 | 0.802 | 0.519 | | |
| GMF + GUFR | 1 | 0.155 | | 0.845 | 0.526 | 0.043 | 0.008 |
| | 2 | 0.198 | | 0.802 | 0.517 | | |
| LightGCN | 1 | 0.212 † | | 0.788 | 0.432 | 0.077 | 0.043 |
| | 2 | 0.289 † | 19 | 0.711 | 0.389 | | |
| LightGCN + GUFR | 1 | 0.128 | | 0.873 | 0.47 | **0.001** | 0.031 |
| | 2 | 0.128 | | 0.872 | 0.44 | | |
| MLP | 1 | 0.173 | | 0.827 | 0.553 | 0.016 | 0.007 |
| | 2 | 0.158 | **7** | 0.842 | 0.56 | | |
| MLP + GUFR | 1 | 0.151 | | 0.849 | 0.557 | 0.014 | **0.004** |
| | 2 | 0.165 | | 0.835 | 0.553 | | |
| NeuMF | 1 | 0.199 | | 0.802 | 0.542 | 0.002 | 0.015 |
| | 2 | 0.198 | 8 | 0.802 | 0.557 | | |
| NeuMF + GUFR | 1 | 0.149 | | 0.851 | 0.556 | 0.05 | 0.005 |
| | 2 | 0.199 | | 0.801 | 0.551 | | |
| Other Fairness Baselines | | | | | | | |
| NFCF | 1 | 0.198 | | 0.802 | 0.539 | 0.01 | 0.01 |
| | 2 | 0.205 † | 8 | 0.795 | 0.549 | | |
| MFCF | 1 | 0.243 † | | 0.757 | 0.552 | 0.002 | 0.009 |
| | 2 | 0.242 † | 9 | 0.758 | 0.561 | | |
| NeuMF-UFR | 1 | 0.216 † | | 0.784 | 0.528 | 0.034 | 0.021 |
| | 2 | 0.182 | 8 | 0.818 | 0.549 | | |
| GMF-UFR | 1 | 0.215 † | | 0.785 | 0.527 | 0.03 | 0.022 |
| | 2 | 0.185 | 9 | 0.815 | 0.549 | | |

**Table 2:** Performances and fairness comparisons with base models and fairness baselines on the **MovieLens Dataset** grouped by the **gender** in terms of risk, average set size, and Hit Rate Diff/NDCG Diff, respectively. Bold indicates the best result, underline indicates the second best and † marks threshold exceeded cases.

The results, presented in Tables 1 to 4 lead us to the following key observations:

- The GUFR framework ensures that all base models generate prediction sets that satisfy both risk control and fairness guarantees across all datasets.

- The GUFR-enhanced models always meet the risk below the defined thresholds. For the base models, the minimum risk threshold criteria is frequently not met. For example, in the AmazonOffice Dataset, we notice, as depicted by †, that the risk thresholds are not met for at least one group, i.e., the disadvantaged group across all the base models. In fairness baselines, we observe the criteria are not met for both the groups in most cases across all the datasets, which may be because of their emphasis on trading off performance for accuracy.

- We also observe that the GUFR-enhanced models can get the best results in average set size on all the datasets, but the best model varies among different datasets. For example, MLP + GUFR achieves the best recommendations in terms of average set size on the AmazonOffice dataset and MovieLens dataset, respectively. Similarly, DeepFM + GUFR returns minimum average set sizes for the Last.fM dataset while GMF + GUFR achieves that for the Book-Crossing dataset, respectively.

**Table 3** (Last.fM Dataset, grouped by Region):

| Method | Group | Risk ↓ | Average Set Size ↓ | Hit Rate | NDCG | Hit Rate Diff ↓ | NDCG Diff ↓ |
|---|---|---|---|---|---|---|---|
| DeepFM | 1 | 0.171 | 29 | 0.829 | 0.363 | 0.108 | 0.111 |
| | 2 | 0.279† | | 0.721 | 0.252 | | |
| DeepFM + GUFR | 1 | 0.181 | | 0.819 | 0.358 | 0.016 | 0.016 |
| | 2 | 0.197 | | 0.803 | 0.342 | | |
| GMF | 1 | 0.186 | 45 | 0.814 | 0.268 | 0.107 | 0.071 |
| | 2 | 0.293† | | 0.707 | 0.197 | | |
| GMF + GUFR | 1 | 0.156 | | 0.844 | 0.273 | 0.019 | 0.023 |
| | 2 | 0.175 | | 0.825 | 0.25 | | |
| LightGCN | 1 | 0.217† | 33 | 0.783 | 0.382 | 0.026 | 0.013 |
| | 2 | 0.243† | | 0.757 | 0.369 | | |
| LightGCN + GUFR | 1 | 0.164 | | 0.836 | 0.392 | 0.03 | 0.02 |
| | 2 | 0.194 | | 0.806 | 0.39 | | |
| MLP | 1 | 0.221† | 32 | 0.779 | 0.328 | 0.019 | 0.013 |
| | 2 | 0.24† | | 0.76 | 0.315 | | |
| MLP + GUFR | 1 | 0.197 | | 0.803 | 0.331 | **0.007** | **0.008** |
| | 2 | 0.19 | | 0.81 | 0.323 | | |
| NeuMF | 1 | 0.201 † | 30 | 0.799 | 0.323 | 0.068 | 0.021 |
| | 2 | 0.269 † | | 0.731 | 0.302 | | |
| NeuMF + GUFR | 1 | 0.187 | | 0.813 | 0.330 | 0.011 | **0.004** |
| | 2 | 0.198 | | 0.802 | 0.326 | | |
| *Other Fairness Baselines* | | | | | | | |
| NFCF | 1 | 0.248 † | 30 | 0.752 | 0.344 | 0.024 | 0.049 |
| | 2 | 0.272† | | 0.728 | 0.295 | | |
| MFCF | 1 | 0.231 † | 45 | 0.769 | 0.269 | 0.066 | 0.051 |
| | 2 | 0.297† | | 0.703 | 0.218 | | |
| NeuMF-UFR | 1 | 0.213 † | 30 | 0.787 | 0.306 | 0.045 | 0.019 |
| | 2 | 0.258 † | | 0.742 | 0.287 | | |
| GMF-UFR | 1 | 0.211 † | 45 | 0.789 | 0.245 | 0.067 | 0.048 |
| | 2 | 0.278 † | | 0.722 | 0.197 | | |

**Table 3:** Performances and fairness comparisons with base models and fairness baselines on the **Last.fM Dataset** grouped by the **Region** in terms of risk, average set size, and Hit Rate Diff/NDCG Diff, respectively. Bold indicates the best result, underline indicates the second best and † marks threshold exceeded cases.

**Table 4** (Book-Crossing Dataset, grouped by Age):

| Method | Group | Risk ↓ | Average Set Size ↓ | Hit Rate | NDCG | Hit Rate Diff ↓ | NDCG Diff ↓ |
|---|---|---|---|---|---|---|---|
| DeepFM | 1 | 0.123 | 39 | 0.873 | 0.291 | 0.302† | 0.115 |
| | 2 | 0.429† | | 0.571 | 0.176 | | |
| DeepFM + GUFR | 1 | 0.188 | | 0.812 | 0.251 | **0.003** | 0.02 |
| | 2 | 0.191 | | 0.809 | 0.231 | | |
| GMF | 1 | 0.187 | 30 | 0.813 | 0.277 | 0.129 | 0.115 |
| | 2 | 0.316† | | 0.684 | 0.162 | | |
| GMF + GUFR | 1 | 0.185 | | 0.815 | 0.268 | 0.116 | 0.05 |
| | 2 | 0.199 | | 0.801 | 0.232 | | |
| LightGCN | 1 | 0.154 | 45 | 0.846 | 0.189 | 0.217† | 0.031 |
| | 2 | 0.371 † | | 0.629 | 0.158 | | |
| LightGCN + GUFR | 1 | 0.18 | | 0.82 | 0.186 | 0.019 | 0.025 |
| | 2 | 0.199 | | 0.801 | 0.161 | | |
| MLP | 1 | 0.124 | 34 | 0.876 | 0.225 | 0.192 | 0.093 |
| | 2 | 0.316† | | 0.684 | 0.132 | | |
| MLP + GUFR | 1 | 0.167 | | 0.833 | 0.194 | 0.029 | **0.019** |
| | 2 | 0.196 | | 0.804 | 0.175 | | |
| NeuMF | 1 | 0.145 | 39 | 0.855 | 0.253 | 0.214† | 0.107 |
| | 2 | 0.359† | | 0.641 | 0.146 | | |
| NeuMF + GUFR | 1 | 0.187 | | 0.813 | 0.227 | 0.004 | 0.036 |
| | 2 | 0.191 | | 0.809 | 0.204 | | |
| *Other Fairness Baselines* | | | | | | | |
| NFCF | 1 | 0.216 † | 39 | 0.784 | 0.264 | 0.095 | 0.08 |
| | 2 | 0.311† | | 0.689 | 0.184 | | |
| MFCF | 1 | 0.248 † | 30 | 0.752 | 0.252 | 0.087 | 0.074 |
| | 2 | 0.335† | | 0.665 | 0.178 | | |
| NeuMF-UFR | 1 | 0.183 | 39 | 0.817 | 0.236 | 0.143 | 0.041 |
| | 2 | 0.326 † | | 0.674 | 0.195 | | |
| GMF-UFR | 1 | 0.195 † | 30 | 0.805 | 0.265 | 0.118 | 0.097 |
| | 2 | 0.313 † | | 0.687 | 0.168 | | |

**Table 4:** Performances and fairness comparisons with base models and fairness baselines on the **Book-Crossing Dataset** grouped by the **Age** in terms of risk, average set size, and Hit Rate Diff/NDCG Diff, respectively. Bold indicates the best result, underline indicates the second best and † marks threshold exceeded cases.

- All GUFR-enhanced models meet the fairness threshold for both the Hit Rate Diff and NDCG Diff across all datasets. However, the best-performing models vary by dataset. For example, MLP + GUFR achieves the best fairness on the AmazonOffice dataset, while LightGCN + GUFR performs best on MovieLens dataset under the Hit Rate Diff; NeUMF + GUFR outperforms all the other models on the Last.fm dataset and MLP + GUFR is superior on the Book-Crossing dataset in terms of NDCG Diff. In addition, the base models do not always achieve the fairness metrics and exceed the fairness threshold marked by †, such as GMF on the AmazonOffice Dataset and LightGCN on the Book-Crossing Dataset. Meanwhile, the fairness baseline models do achieve fairness metrics after sacrificing their accuracy, but they are still inferior to the GUFR-enhanced models.

- Overall, the GUFR framework effectively ensures both recommendation performance and fairness while guaranteeing risk control, providing valuable insights for real-world applications. We further discuss the practical applicability in Appendix A.5

### 6.3.2 PARAMETER ANALYSIS

We further analyze the influence of the pre-defined risk-related parameters $\alpha$ and $\delta$ and fairness-related parameters $\eta$ and $\hat{\delta}$ on the prediction sets generated by GUFR framework.

**Effect of Risk Control Parameters $\alpha$ and $\delta$ on Prediction Set Sizes :** We first evaluate the impact of error rate $\alpha$ varying from 0.10 to 0.50 (in increments of 0.05) on average prediction set sizes under fixed risk confidence thresholds $\delta = 0.05, 0.10, 0.15$ using AmazonOffice dataset, grouped by interactions in Figure 2. It can be easily observed that as $\alpha$ increases, the average set size across all models decreases. The decreasing trend demonstrates the framework's ability to generate valid prediction sets that adapt to the error rate $\alpha$. Similar trends can be observed on remaining datasets, see Figures 6 to 8 in Appendix A.4.1.

We further evaluate the effect of varying risk confidence $\delta$ from 0.10 to 0.50 (in increments of 0.05) on the average prediction set sizes under fixed risk thresholds ($\alpha = 0.15, 0.20, 0.25$) using the Book-Crossing dataset, grouped by age (see Figure 3). In general, all the models show a decreasing trend which validates the effectiveness of the proposed framework. Intrestingly, the prediction set sizes do not to seem to fluctuate much for smaller values of $\delta$ while decreasing trend occurs with increasing $\delta$. This is because relaxing the confidence of the risk constraints makes our predictions less conservative, thereby reducing the number of items included in the prediction set. Similar phenomenon can be obtained on the other datasets, see Figures 9 to 11 in Appendix A.4.1.

**Effect of Fairness Control Parameters $\eta$ and $\hat{\delta}$ on Prediction Set Sizes :** We analyze how varying $\eta$, measured by the Hit Rate Diff. and NDCG Diff. from 0.10 to 0.50 (in increments of 0.05) on the average prediction set

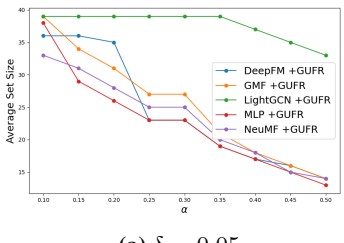 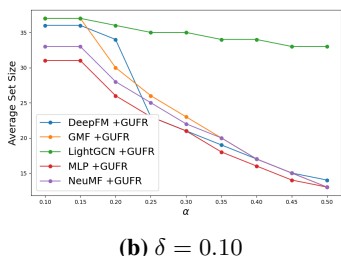 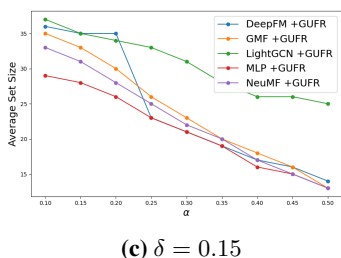

**(a)** $\delta = 0.05$        **(b)** $\delta = 0.10$        **(c)** $\delta = 0.15$

**Figure 2:** Analysis of base models after applying the GUFR framework in terms of average set size with varying $\alpha = \{0.10, 0.15, 0.20, 0.25, 0.30, 0.35, 0.40, 0.45, 0.50\}$ on **AmazonOffice** dataset grouped by **Interactions** under different $\delta$.

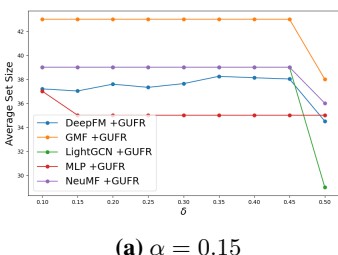 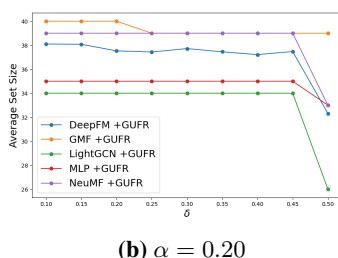 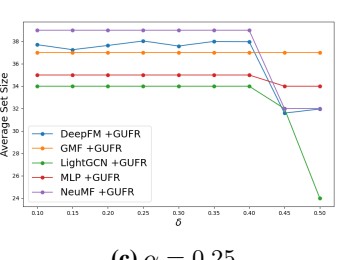

**(a)** $\alpha = 0.15$        **(b)** $\alpha = 0.20$        **(c)** $\alpha = 0.25$

**Figure 3:** Analysis of base models after applying the GUFR framework in terms of average set size with varying $\delta = \{0.10, 0.15, 0.20, 0.25, 0.30, 0.35, 0.40, 0.45, 0.50\}$ on **Book-Crossing** dataset grouped by **Age** under different $\alpha$.

sizes under fixed fairness confidence ($\hat{\delta} = 0.15, 0.20, 0.25$) affects average prediction set sizes, measured on the MovieLens dataset grouped by gender (Figure 4). With increasing $\eta$, the prediction set size decreases, validating model's capacity to have smaller prediction sets for less strict $\eta$ condition. The prediction set sizes usually stabilize after an initial decrease as $\eta$ rises, suggesting that the framework's fairness sensitivity to $\eta$ diminishes beyond a certain point. This offers guidance on selecting appropriate fairness thresholds while maintaining usability. Similar results can be observed on the other datasets, see Figures 12 to 14 in Appendix A.4.1.

Finally, we examine the trends on average prediction set sizes by varying fairness confidence $\hat{\delta}$ from 0.10 to 0.50 (in increments of 0.05) under fixed fairness thresholds ($\eta = 0.15, 0.20, 0.25$) measured on Last.fm dataset grouped by region (Figure 5) . We notice that as the value of $\hat{\delta}$ increases, for a given fairness threshold, the model becomes less conservative and hence prediction set size decreases. This phenomenon further validates the effectiveness of our framework in balancing between producing tight average prediction set size and ensuring fairness. Similarly, results for the other datasets can be found in Figures 15 to 17 in Appendix A.4.1.

To sum up, the above parameter analysis provides a guidance on real-world applications when balance between the recommendation performance and fairness needs to be considered with a confidence guarantee.

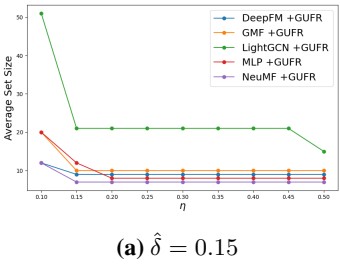 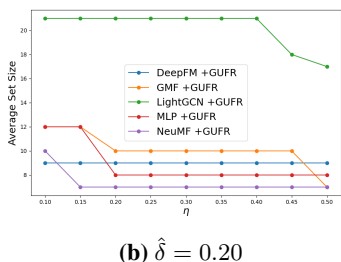 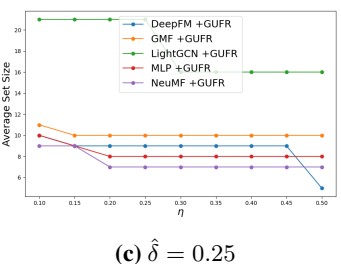

**(a)** $\hat{\delta} = 0.15$        **(b)** $\hat{\delta} = 0.20$        **(c)** $\hat{\delta} = 0.25$

**Figure 4:** Analysis of base models after applying the GUFR framework in terms of average set size with varying $\eta = \{0.10, 0.15, 0.20, 0.25, 0.30, 0.35, 0.40, 0.45, 0.50\}$ on **MovieLens** dataset grouped by **Gender** under different $\hat{\delta}$.

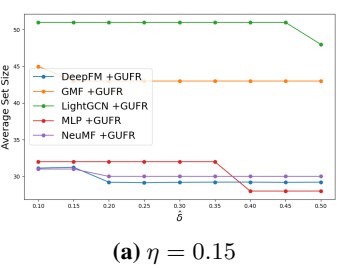 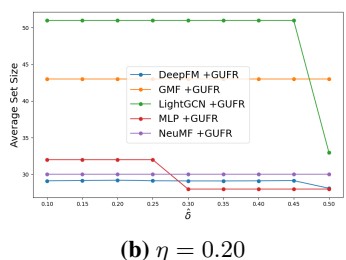 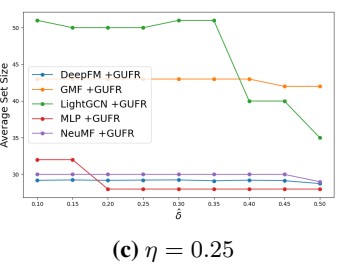

**(a)** $\eta = 0.15$        **(b)** $\eta = 0.20$        **(c)** $\eta = 0.25$

**Figure 5:** Analysis of base models after applying the GUFR framework in terms of average set size with varying $\hat{\delta} = \{0.10, 0.15, 0.20, 0.25, 0.30, 0.35, 0.40, 0.45, 0.50\}$ on **Last.fm** dataset grouped by **Region** under different $\eta$.

| Dataset | MFCF | NFCF | GMF-UFR | NeuMF-UFR | **GUFR** |
|---|---|---|---|---|---|
| AmazonOffice (by interactions) | 45 | 50 | 25 | 22 | **8** |
| MovieLens (by gender) | 75 | 90 | 49 | 45 | **12** |
| Last.fm (by region) | 50 | 58 | 35 | 30 | **8** |
| Book-Crossing (by age) | 110 | 135 | 68 | 65 | **15** |

**Table 5:** Training time (mins) comparison of our framework GUFR with four fairness baselines

### 6.3.3 TIME EFFICIENCY COMPARISON

We analyze the computational cost (training time) of the GUFR framework in comparison with other fairness baselines. Specifically, for in-processing fairness baselines such as NFCF and MFCF, we consider the fine-tuning step to calculate the training time. For post-processinng fairness baselines such as NeuMF-UFR and GMF-UFR, we take the re-reranking step as the training time. For proposed GUFR framework, we take the calibration step as the training time. We measure the time of GUFR, averaged on top of all the base models. Our experiments are conducted via 10-fold cross validation to ensure statistical reliability. The results are presented in Table 5.

From the results, we can observe that our proposed framework GUFR is significantly more time-efficient than the other fairness baselines, which indicates the scalability of our method. This is because in-processing methods like NFCF and MFCF involves model refitting which substantially increases the computational cost. By contrast, GUFR operates independently of the training phase, eliminating this overhead. Additionally, GUFR is substantially faster than NeuMF-UFR and GMF-UFR, other post-processing methods, because these models involve solving a constrained and complex optimization problem, whereas GUFR employs a simple yet effective greedy-based algorithm.

## 7 CONCLUSION

This paper investigates two principle issues that affect the credibility of RS with respect to confidence and fairness. We integrate the two factors into a unified framework called Guaranteed User Fairness in Recommendation (GUFR), which dynamically outputs prediction sets that are guaranteed to have the risk and fairness below a threshold with pre-specified high confidence, such as 90%, while retaining the minimum average size. We conduct theoretical analysis and empirical studies, which are consistent in validating the effectiveness. It is noteworthy that the efficiency of optimizing the GUFR also depends on the tightness of the derived upper bounds for our risk and fairness, thus, we leave the question whether there exists tighter upper bounds for the future work. Moreover, the proposed framework can work on top of any recommendation model by taking them as black-box, which offers a robust foundation for advancing fairness and reliability in RS, paving the way for future research and development in this field.

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

# A APPENDIX

## A.1 ASSUMPTIONS

**Assumption A.1.** *In theorem 2, we assume that the groups $G_1$ and $G_2$ are mutually independent and that hit rates or NDCG scores are independently distributed within each group. Under these assumptions, the variances for the fairness metrics are calculated as follows:*

$$\sigma_{hit}^2 = \frac{\hat{p}_1(1 - \hat{p}_1)}{n_1} + \frac{\hat{p}_2(1 - \hat{p}_2)}{n_2},$$

$$\sigma_{NDCG}^2 = \frac{s_1^2}{n_1} + \frac{s_2^2}{n_2},$$

*where $\hat{p}_1$ and $\hat{p}_2$ are the observed hit rates, and $s_1^2$ and $s_2^2$ are the sample variances of NDCG scores for groups $G_1$ and $G_2$, respectively.*

*This assumption ensures that the application of Bernstein's inequality is valid, allowing us to derive the Upper Confidence Bound (UCB) for fairness metrics as shown in 13.*

**Assumption A.2.** *Throughout the theorem 3, we make a mild assumption on $\lambda^{min}$, i.e., the minimum value the parameter $\lambda$ can take, as follows:*

$$\Pr(R_G(\lambda_G^{min}) \leq \alpha) \geq 1 - \delta \wedge \Pr(\Delta F(\lambda_{G_1}^{min}, \lambda_{G_2}^{min}) \leq \eta) \geq 1 - \delta$$

*where $\lambda_G^{min}$ is the group-specific minimum value of the parameter for risk control, and $\lambda_{G_1}^{min}$ and $\lambda_{G_2}^{min}$ are the minimum values for fairness control across the groups.*

*This assumption depicts the belief that we can control any user-defined risk $\alpha$ and fairness $\epsilon$ by taking valid $\lambda$ values in a closed set $\Lambda \subseteq \mathbb{R}^2 \cup \{\pm\infty\}$.*

## A.2 PROOFS

### A.2.1 PROOF OF THEOREM 1

*Proof.* We focus on finding some $\hat{R}_G^+$ such that out of $n$ samples, $\hat{R}_G$ yields atmost $k = n\hat{R}_G$ successes (where success is defined as observing a risk) with a significance level of atleast $1 - \delta$. The CDF of the binomial distribution is given by:

$$P(\text{Binom}(n, p) \leq k) = \sum_{i=0}^{k} \binom{n}{i} p^i (1 - p)^{n-i}.$$

Let us assume we know $\hat{R}_G^+$ and we seek $\hat{R}_G$ such that:

$$P(\text{Binom}(n, \hat{R}_G^+) \leq n\hat{R}_G) \geq 1 - \delta.$$

Replacing $\hat{R}_G^+$ with the user-defined risk value $\alpha$, the equation becomes:

$$P(\text{Binom}(n, \alpha) \leq n\hat{R}_G) \geq 1 - \delta$$

or

$$P(\text{Binom}(n, \alpha) \leq n\hat{R}_G) \leq \delta$$

which can be reformulated as:

$$\text{BinomCDF}(n\hat{R}_G, n, \alpha) \leq \delta.$$

To solve for $\hat{R}_G$, we find the root of this equation which is also the UCB at $\alpha$: i.e.,

$$\text{BinomCDF}(n\hat{R}_G, n, \alpha) - \delta = 0$$

Formally,

$$\hat{R}_G^+ = \sup \left\{ \hat{R}_G : \text{BinomCDF}(n\hat{R}_G, n, \alpha) \leq \delta \right\}$$

Hence Proved. □

### A.2.2 PROOF OF THEOREM 2

*Proof.* Bernstein's inequality for a sum of independent random variables $X_i$ with mean $\mu$, variance $\sigma^2$, and bounded by $U$ states:

$$P\left(\left|\frac{1}{n}\sum_{i=1}^{n}(X_i - \mu)\right| \geq t\right) \leq 2\exp\left(-\frac{\frac{1}{2}nt^2}{\sigma^2 + \frac{1}{3}Ut}\right), \tag{16}$$

where $n$ is the number of observations, $X_i$ is the $i$-th random variable, $t$ is the deviation threshold, $\sigma^2$ is the variance of $X_i$, and $U$ is the upper bound on the range of $X_i$.

Analogously, we consider, with some decision-maker confidence value $\hat{\delta}$, that the empirical fairness metric differs from the true fairness metric by the threshold $t$. This can be mathematically represented as:

$$\hat{\delta} = 2\exp\left(-\frac{\frac{1}{2}nt^2}{\sigma_F^2 + \frac{1}{3}Ut}\right),$$

which rearranges to:

$$\log\left(\frac{2}{\hat{\delta}}\right) = \frac{\frac{1}{2}nt^2}{\sigma_F^2 + \frac{1}{3}Ut},$$

solving for $t$ gives:

$$t = \sqrt{\frac{2\sigma_F^2 \log\left(\frac{2}{\hat{\delta}}\right) + \frac{2}{3}U\log\left(\frac{2}{\hat{\delta}}\right)}{n}}.$$

Assuming $U = 1$ conservatively and $n = n_1 + n_2$, we obtain the UCB as:

$$\Delta F^+(\lambda_{G_1}, \lambda_{G_2}, \hat{\delta}) = \Delta F(\lambda_{G_1}, \lambda_{G_2}) + \sqrt{\frac{2\sigma_F^2 \log(\frac{2}{\hat{\delta}}) + \frac{2}{3}\log(\frac{2}{\hat{\delta}})}{n_1 + n_2}}.$$

$\square$

### A.2.3 PROOF OF THEOREM 3

*Proof.* Let $\lambda_G^*$ be the highest parameter value for each group $G \in \{G_1, G_2\}$ such that the expected risk of not including truly relevant items and the fairness metric is less than $\alpha$ and $\eta$ respectively, i.e.,

$$\lambda_G^* = \max\{\lambda_G \in [\lambda_{\min,G}, \lambda_{\max,G}] : \\ R_G(\lambda_G) \leq \alpha \wedge \Delta F_G(\lambda_{G_1}, \lambda_{G_2}) \leq \eta\} \tag{17}$$

Assume for a parameter value $\hat{\lambda}_G$, we have $R_G(\hat{\lambda}_G) > \alpha$ or $\Delta F(\hat{\lambda}_{G_1}, \hat{\lambda}_{G_2}) > \alpha$.

Then by the definition of $\lambda_G^*$, we have,

$$R_G(\lambda_G^*) \leq \alpha \wedge \Delta F(\lambda_{G_1}^*, \lambda_{G_2}^*) \leq \eta$$

which implies,

$$R_G(\lambda_G^*) \leq \alpha < R_G(\hat{\lambda}_G) \vee \Delta F(\lambda_{G_1}^*, \lambda_{G_2}^*) \leq \eta < \Delta F(\hat{\lambda}_{G_1}, \hat{\lambda}_{G_2})$$

Given that the risk $R_G$ and fairness $\Delta F$ are monotonic, we conclude,

$$\hat{\lambda}_G > \lambda_G^*$$

Since $\hat{\lambda}_G$ and $\lambda_G^*$ are within the range of real numbers, consider some $\xi > 0$ such that

$$(\lambda_G^* + \xi) \geq \hat{\lambda}_G,$$

Utilizing the definition of $\lambda_G^*$ and $\hat{\lambda}_G$ in Equation 17, we get,

$$R_G^+(\lambda_G^* + \xi, \delta) \leq \alpha < R_G(\lambda_G^* + \xi) \\ \vee \Delta F^+(\lambda_{G_1}^*, \lambda_{G_2}^* + \xi, \delta) \leq \eta < \Delta F(\lambda_{G_1}^*, \lambda_{G_2}^* + \xi) \tag{18}$$

According to the principles of Upper Confidence Bound (UCB), i.e., eq. 10, the events $R_G^+(\lambda_G^* + \xi, \delta) \leq \alpha$ or $\Delta F^+(\lambda_{G_1}^*, \lambda_{G_2}^* + \xi, \hat{\delta}) \leq \eta$ can only occur with probabilities not exceeding $\delta$ and $\hat{\delta}$ respectively. Specifically, the UCB ensures that the probability of observing $R_G(\hat{\lambda}_G) > \alpha$ is bounded by $\delta$, or the probability of $\Delta F(\hat{\lambda}_{G_1}, \hat{\lambda}_{G_2}) > \eta$ is bounded by $\hat{\delta}$.

Therefore, with complementary probability condition, under Assumption 1 and the defined ranges of $\delta$ and $\hat{\delta}$, we can conclude with confidence that:

$$\Pr(R_G(\hat{\lambda}_G) \leq \alpha) \geq 1 - \delta \quad \wedge$$
$$\Pr(\Delta F(\hat{\lambda}_{G_1}, \hat{\lambda}_{G_2}) \leq \eta) \geq 1 - \hat{\delta}. \tag{19}$$

This validates the assertions of Theorem 3, thereby formally proving the theorem.

$\square$

### A.2.4 PROOF OF THEOREM 4

*Proof.* Since $R_G(\phi_{\lambda_{*,G}}) \leq R_G(\phi_{\hat{\lambda}_G})$ and $\Delta F(\phi_{\lambda_{G_1}, \lambda_{G_2}^*}) \leq \Delta F(\phi_{\lambda_{G_1}, \hat{\lambda}_{G_2}})$, this relationship is expressed through the sum of relevance scores $m(u, i)$ over the items in the respective prediction sets for users:

$$\sum_{u \in G} \sum_{i \in \phi_{\lambda_G^*}(u)} m(u, i) \geq \sum_{u \in G} \sum_{i \in \phi_{\hat{\lambda}_G}(u)} m(u, i),$$

indicating that the accumulated scores of included items in $\phi_{\lambda_G^*}$ are greater.

This is equivalent to:

$$\sum_{u \in G} \sum_{i \in \phi_{\lambda_G^*}(u) \setminus \phi_{\hat{\lambda}_G}(u)} m(u, i) \geq \sum_{u \in G} \sum_{i \in \phi_{\hat{\lambda}_G}(u) \setminus \phi_{\lambda_G^*}(u)} m(u, i).$$

For some items $i \in \phi_{\lambda_G^*}(u) \setminus \phi_{\hat{\lambda}_G}(u)$, $m(u, i) < \hat{\lambda}_G$, and for all items $i \in \phi_{\hat{\lambda}_G}(u) \setminus \phi_{\lambda_G^*}(u)$, $m(u, i) \geq \hat{\lambda}_G$, based on Algorithm 1.

This condition is satisfied if:

$$|\phi_{\lambda_G^*}(u)| \geq |\phi_{\hat{\lambda}_G}(u)|.$$

Thus, the expected size of the set using $\phi_{\hat{\lambda}_G}$ is optimized to be minimal, i.e.,

$$E\left[|\phi_{\hat{\lambda}_G}(u)|\right] \leq E\left[|\phi_{\lambda_G^*}(u)|\right], \tag{20}$$

thereby proving the theorem.

$\square$

## A.3 DETAILED EXPERIMENTATION DETAILS

### A.3.1 DATASETS AND GROUPING METHODS

In the main paper, we introduced four user grouping strategies to evaluate the fairness and performance of our framework: (1) grouping based on interaction count with items, (2) grouping based on user age, (3) grouping based on user gender, and (4) grouping based on geographic categorization into developed and other countries. These strategies were applied to the AmazonOffice, Book-Crossing, MovieLens, and Last.fm datasets, respectively. Below, we provide further details on the grouping methodology:

- **Grouping by interaction count:** Following Li et al. (2021a), users were initially evenly split into two groups, with 50% assigned to each group. The groups were then dynamically adjusted to ensure that the minimum interaction count in the advantaged group exceeded the maximum count in the disadvantaged group by at least one.

- **Grouping by age:** Users were divided into two age groups: younger users ($\leq 60$ years) and older users ($> 60$ years).

- **Grouping by gender:** Users were grouped into binary categories based on identified gender (male and female).

- **Geographic categorization:** Users were categorized based on their country of origin into developed (e.g., USA, UK, Europe, Japan etc.) and other countries.

Furthermore, we conducted an additional grouping experiment on the Last.fm dataset. We extended the interaction count-based grouping to incorporate interactions with popular items, following Abdollahpouri et al. (2019). The results of this experiment are provided in Appendix A.4.2.

### A.3.2 Sampling and Data Splitting

We followed the following sampling and splitting method:

- **Negative sampling:** Following Ma et al. (2024), we selected 50 non-interacted items per user through negative sampling for training, validation, and testing.

- **Data splitting:** We employed the Leave-One-Out (LOO) strategy (He et al., 2017; Han et al., 2023) to partition the dataset into training, calibration, and testing sets. Specifically, for each user, one interaction was isolated for calibration and testing, while the remaining interactions were used for training.

- **Multiple trials:** To account for variability in sampling and splitting, we repeated the experiments over 20 independent trials. For each trial, random negative samples were drawn for training, validation, and testing. The results were averaged across all the trials.

### A.3.3 Model Configurations and Fairness Baselines

To evaluate the effectiveness of our framework, we implemented it on top of the five base recommender models specified in the main paper. Here, we provide specific architectural and training details of the models used:

**Base Recommendation Models**

- **DeepFM:** Combines 8 latent factors with deep layers of [50, 25, 10] and ReLU activation.
- **GMF:** Utilizes an embedding size of 8 for capturing linear interactions between user and item embeddings.
- **MLP:** Employs layers of [64, 32, 16] with ReLU activation for modeling non-linear interactions.
- **NeuMF:** Integrates GMF and MLP with a GMF embedding size of 8 and MLP layers of [64, 32, 16], using ReLU activation.
- **LightGCN:** Configured with an embedding size of 8 and 3 graph convolution layers.

To validate our framework further, we compared it with four fairness baseline approaches. The baselines are based on the most commonly adopted methods in fairness literature i.e. in-processing and post-processing methods (Li et al., 2023) :

**Fairness Baselines**

- **NFCF and MFCF(In-processing) (Islam et al., 2021):** The authors utilize a pre-training and fine-tuning approach to induce user-sided group fairness. Initially, the user embeddings are learned from non-sensitive interactions, followed by a de-biasing step to mitigate the embedding bias. Finally, the models are fine-tuned on sensitive item recommendations with a fairness penalty to reduce systemic bias in predictions.

- **Neumf-UFR AND GMF-UFR (Post-processing) Li et al. (2021a):** This post-hoc re-ranking approach utilizes an integer programming solver to balance fairness and utility disparity between advantaged and disadvantaged user groups. The method optimizes preference scores while enforcing a fairness constraint, ensuring that recommendation quality differences (e.g., NDCG@10, F1@10) between groups remain below a specified threshold.

### A.4 Additional Experiments

### A.4.1 Parameters Analysis -Continued

**Effect of Risk Control Parameters $\alpha$ and $\delta$ on Prediction Set Sizes**.

Figures 6 to 8 illustrate the trends in the average prediction set size as $\alpha$ varies from 0.10 to 0.50 (in increments of 0.05), while keeping the risk confidence thresholds fixed at $\delta = 0.05, 0.10, 0.15$, using the Book-Crossing, MovieLens, and Last.fm datasets respectively. Similarly, Figures 9 to 11 present the trends in the average prediction set size as $\delta$ varies from 0.10 to 0.50 (in increments of 0.05), while keeping the confidence thresholds fixed at $\alpha = 0.15, 0.20, 0.25$, using the AmazonOffice, MovieLens, and Last.fm datasets respectively.

The observed trends in Figures 6 to 8 (variation in $\alpha$) and Figures 9 to 11 (variation in $\delta$) are consistent with the observations reported in Figure 2 (AmazonOffice dataset) and Figure 3 (Book-Crossing dataset) in the main paper. These results reinforce the consistency of our framework's behavior across different datasets and grouping methods.

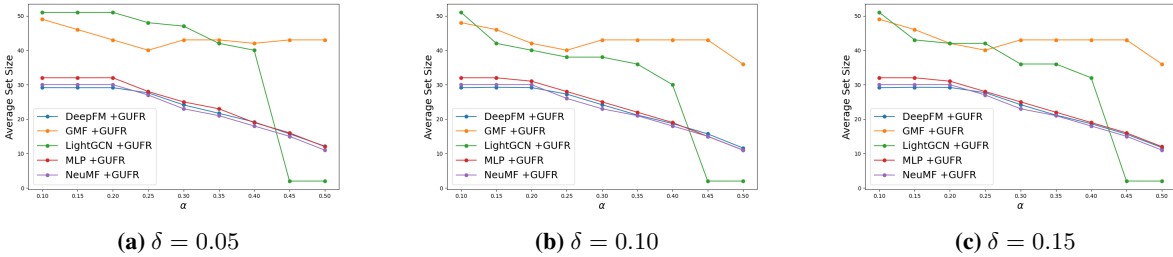

**(a)** $\delta = 0.05$  **(b)** $\delta = 0.10$  **(c)** $\delta = 0.15$

**Figure 6:** Analysis of base models after applying the GUFR framework in terms of average set size with varying $\alpha = \{0.10, 0.15, 0.20, 0.25, 0.30, 0.35, 0.40, 0.45, 0.50\}$ on **Last.fm** dataset grouped by **Region** under different $\delta$.

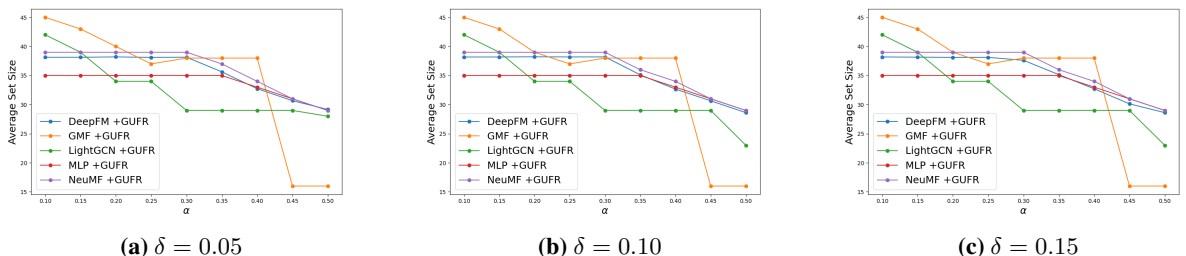

**(a)** $\delta = 0.05$  **(b)** $\delta = 0.10$  **(c)** $\delta = 0.15$

**Figure 7:** Analysis of base models after applying the GUFR framework in terms of average set size with varying $\alpha = \{0.10, 0.15, 0.20, 0.25, 0.30, 0.35, 0.40, 0.45, 0.50\}$ on **Book-Crossing** dataset grouped by **Age** under different $\delta$.

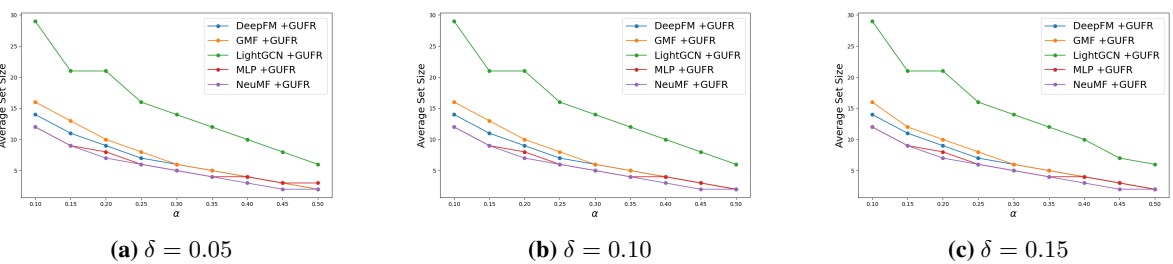

**(a)** $\delta = 0.05$  **(b)** $\delta = 0.10$  **(c)** $\delta = 0.15$

**Figure 8:** Analysis of base models after applying the GUFR framework in terms of average set size with varying $\alpha = \{0.10, 0.15, 0.20, 0.25, 0.30, 0.35, 0.40, 0.45, 0.50\}$ on **MovieLens** dataset grouped by **Gender** under different $\delta$.

**Effect of Fairness Control Parameters $\eta$ and $\hat{\delta}$ on Prediction Set Sizes** Figures 12 to 14 illustrate how the average prediction set size changes as $\eta$ varies from 0.10 to 0.50 (in increments of 0.05), while holding the fairness confidence thresholds fixed at $\hat{\delta} = 0.15, 0.20, 0.25$. These results are based on the AmazonOffice dataset, Book-Crossing and Book-Crossing datasets respectively.

In contrast, Figures 15 to 17 display the trends in prediction set size as $\hat{\delta}$ ranges from 0.10 to 0.50 (in increments of 0.05), with fixed thresholds of $\eta = 0.15, 0.20, 0.25$. These findings are based on AmazonOffice, MovieLens and Last.fm datasets respectively.

These results further validate variations in $\eta$ and $\hat{\delta}$ exhibit consistent patterns, emphasizing our framework's ability to adapt prediction set sizes effectively based on fairness constraints.

### A.4.2 GENERALIZABLITY OF GROUPING METHODS

We validate if GUFR is adaptable to practitioners' demands for customized user groups based on specific biases or fairness concerns relevant to their context. Specifically, we test our framework using different grouping techniques on

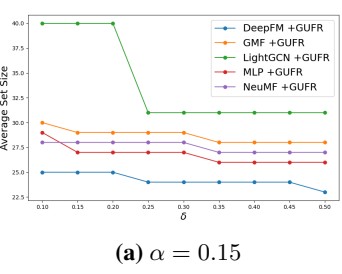
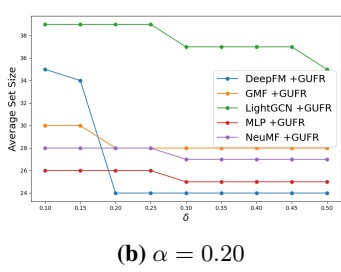
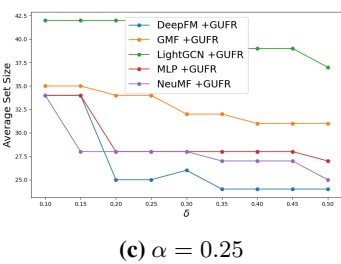

**(a)** $\alpha = 0.15$        **(b)** $\alpha = 0.20$        **(c)** $\alpha = 0.25$

**Figure 9:** Analysis of base models after applying the GUFR framework in terms of average set size with varying $\delta = \{0.10, 0.15, 0.20, 0.25, 0.30, 0.35, 0.40, 0.45, 0.50\}$ on **AmazonOffice** dataset grouped by **Interactions** under different $\alpha$.

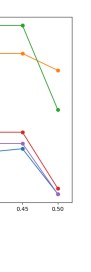
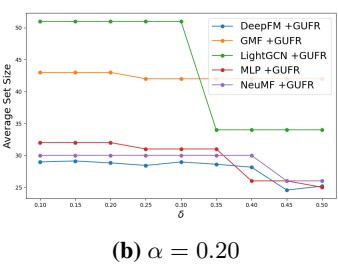
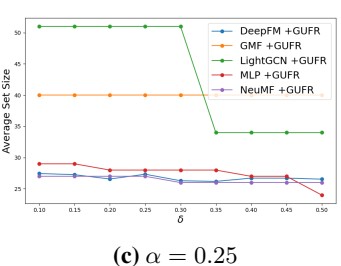

**(a)** $\alpha = 0.15$        **(b)** $\alpha = 0.20$        **(c)** $\alpha = 0.25$

**Figure 10:** Analysis of base models after applying the GUFR framework in terms of average set size with varying $\delta = \{0.10, 0.15, 0.20, 0.25, 0.30, 0.35, 0.40, 0.45, 0.50\}$ on **Last.fm** dataset grouped by **Region** under different $\alpha$.

a single dataset i.e. Last.fm by grouping users based on item interactions and grouping by both item interactions and interactions with popular items on the Last.fm dataset. The results could be found in Table 6 and Table 7. The results demonstrate that the GUFR framework can dynamically generate prediction sets for users grouped by any condition. This is particularly useful in real-world scenarios, where different applications may have different definitions of fairness. By allowing any grouping method, the framework can support dynamic fairness criteria that can evolve with changing societal norms or organizational policies, thereby allowing practitioners to define user groups based on the specific biases or fairness concerns relevant to their context.

## A.5 PRACTICAL APPLICABILITY OF THE FRAMEWORK

We now analyze the practical applicability of our framework. In real-world recommendation systems, prediction sets are often fixed to a specific size $k$ and applied uniformly across all users. This fixed size is typically determined heuristically or through trial and error, aiming to maximize the likelihood of including items that users may interact with while prioritizing and ranking items by relevance. However, this heuristic approach has several limitations:

- Fixed-size sets can lead to cognitive overload for users when the size is too large or fail to meet individual user needs when the size is too small.

- They do not account for disparities in user engagement or group fairness, potentially disadvantaging certain user groups.

- Recommending unnecessary items results in resource inefficiencies for platforms.

Our framework addresses these challenges by dynamically determining the minimum prediction set size for each user, satisfying fairness and performance guarantees with statistical confidence (e.g., 95%). This complements the existing recommender systems as we can employ an appropriate aggregation method (for example, mean) to compute global k. This global k, obtained with the theoretical guarantees, can then be applied to unseen users, ensuring that fairness and performance guarantees hold across the system.

For example, in e-commerce platforms such as Amazon, instead of heuristically fixing $k = 10$ for all users, our framework identifies an optimal $k$ (e.g., $k = 7$) that balances fairness and accuracy, reducing unnecessary recommendations and enhancing user satisfaction while optimizing platform resources. Similarly, in streaming services like Netflix, dynamically adjusting $k$ in cold-start scenarios ensures concise and personalized recommendations, preventing user overwhelm and aligning with platform resource constraints.

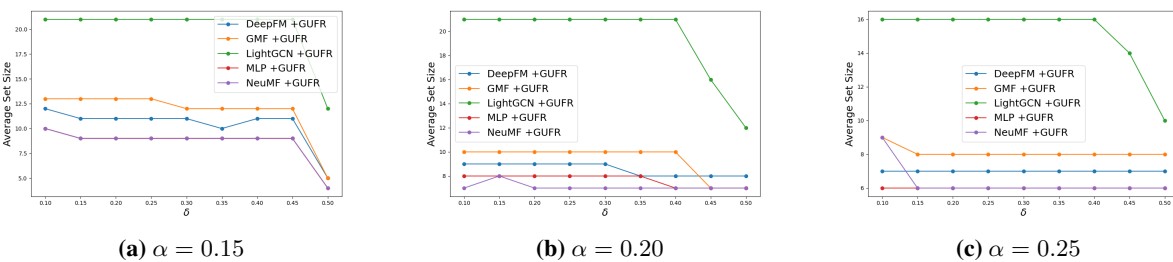

**(a)** $\alpha = 0.15$      **(b)** $\alpha = 0.20$      **(c)** $\alpha = 0.25$

**Figure 11:** Analysis of base models after applying the GUFR framework in terms of average set size with varying $\delta = \{0.10, 0.15, 0.20, 0.25, 0.30, 0.35, 0.40, 0.45, 0.50\}$ on **MovieLens** dataset grouped by **Gender** under different $\alpha$.

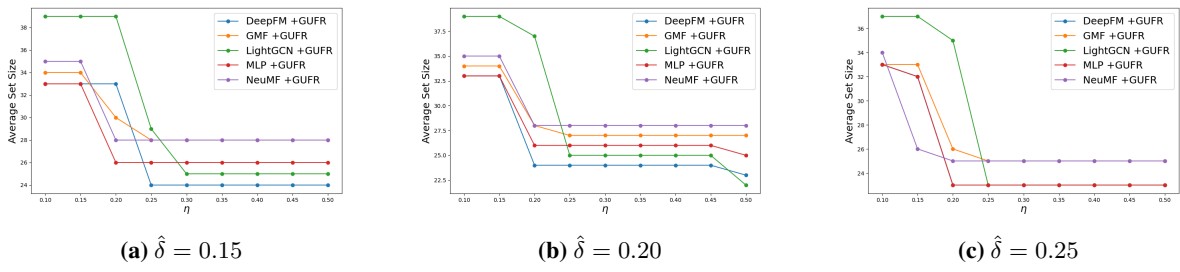

**(a)** $\hat{\delta} = 0.15$      **(b)** $\hat{\delta} = 0.20$      **(c)** $\hat{\delta} = 0.25$

**Figure 12:** Analysis of base models after applying the GUFR framework in terms of average set size with varying $\eta = \{0.10, 0.15, 0.20, 0.25, 0.30, 0.35, 0.40, 0.45, 0.50\}$ on **AmazonOffice** dataset grouped by **Interactions** under different $\hat{\delta}$.

Additionally, the calculated $k$ can serve as a benchmark to fine-tune recommendation models, enabling iterative improvements that enhance fairness and accuracy across diverse user groups. By tailoring prediction set sizes dynamically, our framework provides a practical, scalable solution for modern recommendation systems.

.

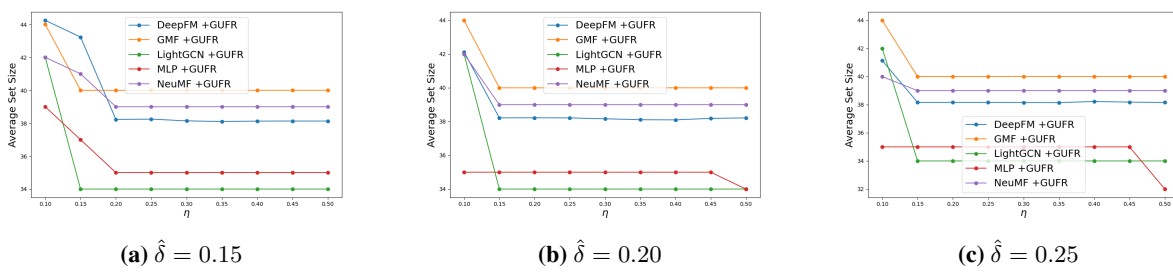

**Figure 13:** nalysis of base models after applying the GUFR framework in terms of average set size with varying $\eta = \{0.10, 0.15, 0.20, 0.25, 0.30, 0.35, 0.40, 0.45, 0.50\}$ on **Book-Crossing** dataset grouped by **Age** under different $\hat{\delta}$.

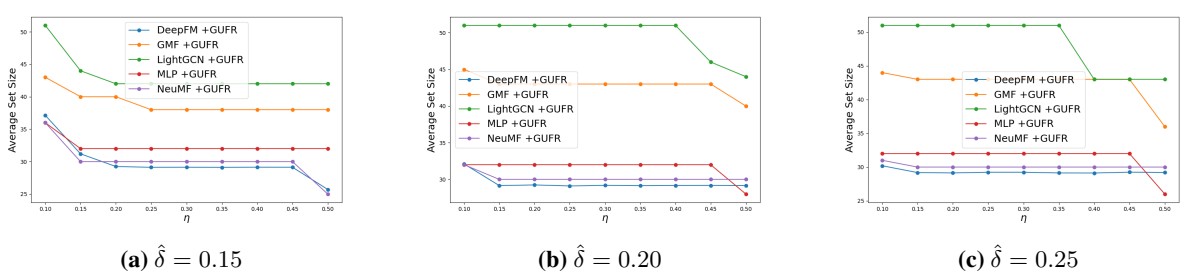

**Figure 14:** Analysis of base models after applying the GUFR framework in terms of average set size with varying $\eta = \{0.10, 0.15, 0.20, 0.25, 0.30, 0.35, 0.40, 0.45, 0.50\}$ on **Last.fM** dataset grouped by **Region** under different $\hat{\delta}$.

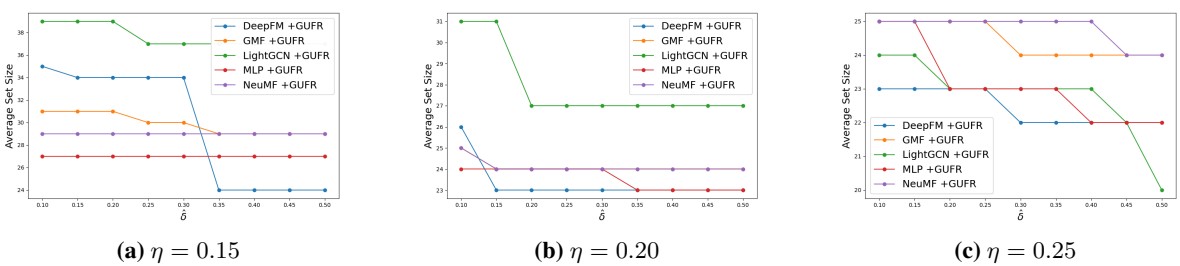

**Figure 15:** Analysis of base models after applying the GUFR framework in terms of average set size with varying $\hat{\delta} = \{0.10, 0.15, 0.20, 0.25, 0.30, 0.35, 0.40, 0.45, 0.50\}$ on **AmazonOffice** dataset grouped by **Interactions** under different $\eta$.

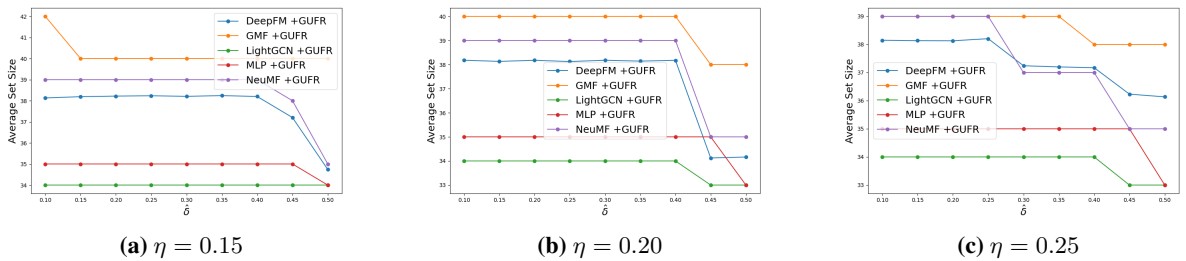

**Figure 16:** Analysis of base models after applying the GUFR framework in terms of average set size with varying $\hat{\delta} = \{0.10, 0.15, 0.20, 0.25, 0.30, 0.35, 0.40, 0.45, 0.50\}$ on **Book-Crossing** dataset grouped by **Age** under different $\eta$.

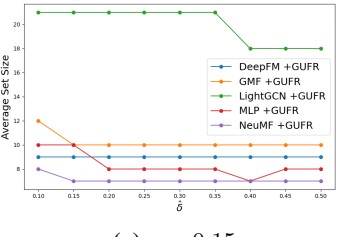 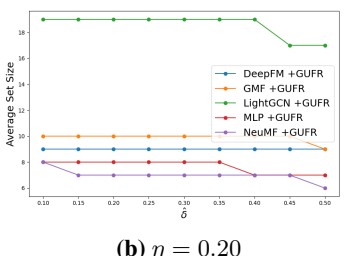 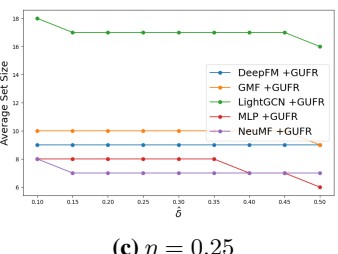

**(a)** $\eta = 0.15$      **(b)** $\eta = 0.20$      **(c)** $\eta = 0.25$

**Figure 17:** Analysis of base models after applying the GUFR framework in terms of average set size with varying $\hat{\delta} = \{0.10, 0.15, 0.20, 0.25, 0.30, 0.35, 0.40, 0.45, 0.50\}$ on **MovieLens** dataset grouped by **Gender** under different $\eta$.

| Method | Group | Risk↓ | Average Set Size ↓ | Hit Rate | NDCG | Hit Rate Diff↓ | NDCG Diff↓ |
|---|---|---|---|---|---|---|---|
| **Grouped by number of interactions** | | | | | | | |
| DeepFM | 1 | 0.183 | 16 | 0.817 | 0.494 | 0.073 | 0.035 |
| | 2 | 0.255 | | 0.745 | 0.458 | | |
| DeepFM + GUFR | 1 | 0.177 | | 0.823 | 0.494 | 0.013 | 0.022 |
| | 2 | 0.19 | | 0.81 | 0.472 | | |
| GMF | 1 | 0.163 | 13 | 0.837 | 0.567 | 0.08 | 0.066 |
| | 2 | 0.243 † | | 0.757 | 0.501 | | |
| GMF + GUFR | 1 | 0.183 | | 0.817 | 0.56 | **0.001** | 0.045 |
| | 2 | 0.183 | | 0.817 | 0.515 | | |
| LightGCN | 1 | 0.179 | **12** | 0.821 | 0.547 | 0.18 | 0.089 |
| | 2 | 0.359† | | 0.641 | 0.458 | | |
| LightGCN + GUFR | 1 | 0.201 | | 0.799 | 0.492 | 0.003 | 0.066 |
| | 2 | 0.198 | | 0.802 | 0.426 | | |
| MLP | 1 | 0.192 | 15 | 0.808 | 0.44 | 0.077 | 0.076 |
| | 2 | 0.269 † | | 0.731 | 0.364 | | |
| MLP + GUFR | 1 | 0.151 | | 0.849 | 0.448 | 0.041 | 0.067 |
| | 2 | 0.192 | | 0.808 | 0.38 | | |
| NeuMF | 1 | 0.151 | 16 | 0.849 | 0.58 | 0.087 | 0.081 |
| | 2 | 0.238 † | | 0.762 | 0.499 | | |
| NeuMF + GUFR | 1 | 0.142 | | 0.858 | 0.581 | 0.027 | 0.067 |
| | 2 | 0.169 | | 0.831 | 0.513 | | |
| **Other Fairness Baselines** | | | | | | | |
| NFCF [a] | 1 | 0.248 † | 16 | 0.822 | 0.569 | 0.039 | 0.053 |
| | 2 | 0.272† | | 0.783 | 0.516 | | |
| MFCF [a] | 1 | 0.231 † | 13 | 0.815 | 0.529 | 0.042 | 0.021 |
| | 2 | 0.297† | | 0.773 | 0.508 | | |
| NeuMF-UFR [b] | 1 | 0.213 † | 16 | 0.827 | 0.546 | 0.045 | 0.031 |
| | 2 | 0.258 † | | 0.782 | 0.515 | | |
| GMF-UFR [b] | 1 | 0.211 † | 13 | 0.819 | 0.536 | 0.047 | **0.019** |
| | 2 | 0.278 † | | 0.772 | 0.517 | | |

**Table 6:** Performance and fairness comparisons with base models and fairness baselines on the **Last.fM Dataset** grouped by the **Item Interactions** in terms of risk, average set size, and Hit Rate Diff/NDCG Diff, respectively. Bold indicates the best result, underline indicates the second best and † marks threshold exceeded cases.

| Method | Group | Loss | Average Set Size | Hit Rate | NDCG | Hit Rate Diff | NDCG Diff |
|---|---|---|---|---|---|---|---|
| **Grouped by number of total interactions & popular items interactions** | | | | | | | |
| DeepFM | 1 | 0.078 | 30 | 0.922 | 0.56 | 0.22† | 0.226 † |
| | 2 | 0.298† | | 0.702 | 0.334 | | |
| DeepFM + GUFR | 1 | 0.162 | | 0.838 | 0.54 | 0.162 | 0.151 |
| | 2 | 0 | | 1 | 0.389 | | |
| GMF | 1 | 0.063 | 30 | 0.937 | 0.603 | 0.112 | 0.231 |
| | 2 | 0.176 | | 0.824 | 0.372 | | |
| GMF + GUFR | 1 | 0.163 | | 0.837 | 0.578 | 0.163 | 0.174 |
| | 2 | 0 | | 1 | 0.405 | | |
| LightGCN | 1 | 0.076 | 45 | 0.924 | 0.627 | 0.119 | 0.24† |
| | 2 | 0.195 | | 0.805 | 0.387 | | |
| LightGCN + GUFR | 1 | 0.126 | | 0.874 | 0.585 | **0.03** | 0.106 |
| | 2 | 0.156 | | 0.844 | 0.479 | | |
| MLP | 1 | 0.057 | 31 | 0.943 | 0.522 | 0.152 | 0.239 † |
| | 2 | 0.209† | | 0.791 | 0.283 | | |
| MLP + GUFR | 1 | 0.143 | | 0.857 | 0.5 | 0.143 | 0.179 |
| | 2 | 0 | | 1 | 0.322 | | |
| NeuMF | 1 | 0.07 | **29** | 0.93 | 0.661 | 0.147 | 0.263 † |
| | 2 | 0.217 † | | 0.783 | 0.398 | | |
| NeuMF + GUFR | 1 | 0.154 | | 0.846 | 0.64 | 0.154 | 0.198 |
| | 2 | 0 | | 1 | 0.442 | | |
| **Other Fairness Baselines** | | | | | | | |
| NFCF | 1 | 0.115 | 29 | 0.885 | 0.629 | 0.08 | 0.208† |
| | 2 | 0.195 | | 0.805 | 0.421 | | |
| MFCF | 1 | 0.137 | 30 | 0.863 | 0.549 | 0.109 | **0.051** |
| | 2 | 0.243† | | 0.757 | 0.44 | | |
| NeuMF-UFR | 1 | 0.111 | 29 | 0.889 | 0.588 | 0.077 | 0.166 |
| | 2 | 0.188 | | 0.812 | 0.428 | | |
| GMF-UFR | 1 | 0.121 | 30 | 0.879 | 0.566 | 0.057 | 0.198 |
| | 2 | 0.178 | | 0.822 | 0.368 | | |

**Table 7:** Performance, and fairness comparisons with base models and fairness baselines on the **Last.fM Dataset** grouped by the **Item Interactions & Interaction with Popular Items** in terms of risk, average set size, and Hit Rate Diff/NDCG Diff, respectively. Bold indicates the best result, underline indicates the second best and † marks threshold exceeded cases.

