# OpenReview forum: "GUARANTEED USER FAIRNESS IN RECOMMENDATION"
_ICLR.cc/2025/Conference — ICLR 2025 Conference Withdrawn Submission_

### Official Review · Reviewer_NF9U · 2024-10-16

**Soundness:** 2
**Presentation:** 2
**Contribution:** 2
**Rating:** 6
**Confidence:** 3

**Summary:**

The paper introduces a framework named GUFR, which aims to generate high-confidence recommendation sets satisfying fairness criteria. The authors provide statistical guarantees, and the results on two public datasets demonstrate the effectiveness of proposed method.

**Strengths:**

S1. The authors provide theoretical analysis.

S2. The experimental results validate that the proposed framework enables various baseline models to meet the defined fairness conditions and risk control.

**Weaknesses:**

W1. Why the fairness metric is defined as the difference in recommendation performance (e.g., NDCG) between two groups requires further discussion. For instance, if Group G1 and Group G2 consist of males and females, respectively, recommending gender-preferred items to each group might achieve similar recommendation performance, but such recommendations could be discriminatory based on gender, which is unfair. In my view, the balanced groups (not limited to just gender) seem to be an important underlying assumption, which is not discussed.

W2. The objective function of minimizing the size of the recommendation set is puzzling. While a smaller set might help reduce uncertainty in recommendations, this conflicts with the objective of encouraging users to purchase or click as much as possible.

**Questions:**

Q1. Why does the constrained optimization problem defined by equation (7) exist a solution? It seems that requiring all users to satisfy the risk and fairness constraints does not necessarily guarantee the existence of a solution.

Q2. Is the parameter $\lambda$ (defined in line 144-145) a scalar on $\mathbb{R}$? What is $\Delta$ in the update formula for parameter $\lambda$ in line 11 of Algorithm 1, and why can such update formula obtain the **optimal** $\lambda$ that satisfies the required conditions.

---

> ### Author Response · Authors · 2024-11-24
>
> We sincerely thank the reviewer for their thoughtful feedback and valuable suggestions. Below, we address each concern and question raised.
>
> ---
>
> ## Why is the fairness metric defined as the difference in recommendation performance (e.g., NDCG) between groups? Could this lead to discriminatory recommendations, particularly with gender-based preferences?
>
> ### Answer:
> We understand the reviewer’s valid concern regarding fairness metrics based on group performance differences. However, achieving similar performance across groups is not inherently discriminatory if the recommendations reflect actual user preferences. For example, if data shows that male users predominantly prefer action novels while female users prefer romance novels, tailoring recommendations to these preferences enhances user satisfaction and reflects personalization, not discrimination.
>
> We believe discrimination arises when recommendations enforce stereotypes, such as assuming all males prefer action novels, thereby excluding individuals with differing preferences. Our framework ensures fairness by tailoring recommendations based on individual preferences within groups, rather than generalizing or enforcing stereotypes. Therefore, our approach does not lead to such discriminatory recommendations.
>
> ---
>
> ## Why does the objective of minimizing recommendation set size conflict with encouraging user engagement (e.g., clicks or purchases)?
>
> ### Answer:
> We acknowledge and agree with the reviewer that user engagement is a critical goal for recommender systems. However, concise recommendation sets can enhance engagement by reducing cognitive load on users as they are presented with only highly relevant items. This approach minimizes decision-making burden, improves user satisfaction, and fosters trust in the system. Additionally, from the platform’s perspective, compact sets optimize computational efficiency, making the system scalable for large-scale applications while maintaining user retention.
>
> Although larger sets may increase the volume of items displayed, they risk overwhelming users and reducing the quality of engagement (e.g., click or purchase of an item). Our method strikes a balance by ensuring high-confidence, targeted recommendations that support usability and improve engagement.
>
> ---
>
> ## Why does the constrained optimization problem in equation (7) guarantee a solution?
>
> ### Answer:
> We thank the reviewer for raising this important question. The existence of a solution for Eq. (7) in our framework is supported by both theoretical and empirical evidence. Theoretically, we utilize Theorems 1 and 2 to provide upper-bound constraints for risk and fairness metrics, while Theorem 3 guarantees the feasibility of solutions through concentration inequalities. These theorems are detailed in Sections 4 and 5, with proofs in Appendix A2.
>
> Algorithm 1 iteratively adjusts group-specific parameters ($\lambda_{G1}$ and $\lambda_{G2}$) to align empirical risk and fairness bounds with pre-defined thresholds, ensuring convergence. Experiments on diverse datasets demonstrate consistent solution feasibility, even with varying group characteristics, as discussed in Section 6 (highlighted in red).
>
> ---
>
> ## What is the parameter $\lambda$ in line 144-145, and how does the update formula in Algorithm 1 ensure an optimal solution?
>
> ### Answer:
> The parameter $\lambda$ is a scalar in $[0,1]$, representing the confidence threshold for including relevant items in the prediction set. The update formula in Algorithm 1 iteratively adjusts $\lambda$ to balance prediction set size, risk control, and fairness constraints.
>
> $\Delta$, the step size used in the updates, is manually validated to ensure an appropriate balance between convergence speed and prediction set size.
>
> As shown in Theorem 3, the updated $\lambda$ satisfies the required fairness and risk conditions. We further utilize Theorem 4 to guarantee that the final prediction set size is minimized without violating risk and fairness guarantees. Together, these elements provide theoretical support for the parameter update mechanism in Algorithm 1.
>
> ---
>
> We sincerely appreciate the reviewer’s insights and comments. Thank you for your time and consideration.

---

> ### Comment · Reviewer_NF9U · 2024-11-25
> **The problem formulation is not appropriate and learning algorithm requires more analysis, and I will maintain the my score.**
>
> Thanks for your rebuttal, but some questions require further discussion.
>
> **Regarding Problem Definition:**
>
> The objectives of this paper can be summarized in two aspects:
>
> - To generate the minimal item prediction set for each user, ensuring it covers their true preferences with high confidence.
> - At the same time, to guarantee comparable recommendation performance (e.g., purchase rate) between two known demographic groups.
>
> The concept of ‘user fairness’ proposed in this paper differs from the commonly used concept of ‘fairness’ for protecting sensitive attributes. Instead, the fairness here is intended to ensure prediction **stability**. Therefore, the use of the term "fairness" may not be entirely appropriate.
>
> **Regarding the Learning Algorithm:**
>
> Is it possible to explicitly write out the objective function of $\lambda = (\lambda_{G1}, \lambda_{G2})$? Given the current definition of the 0-1 loss (Equation (1)), constraints such as NDCG or HR, and the objective of minimizing the size of the prediction set, all these constitute a non-convex optimization problem. Although the authors derive upper bounds and demonstrate the existence of an optimal value, **this does not imply the existence of an optimal solution**. Moreover, even if an optimal solution exists, **there is no guarantee that it can be practically found**.
>
> Based on these two weaknesses, I will maintain the my score.

---

> ### Author Response · Authors · 2024-11-26
>
> We thank the reviewer for their  detailed feedback thoughtful response. Below, we try to address their concerns:
> ### **Question**  The concept of ‘user fairness’ ...
> ### Answer
> We thank the reviewer for raising this point. Our fairness metric builds on well-known works in user-sided group fairness [1] and uses $\Delta F(\lambda_{G_1}, \lambda_{G_2})$ to measure performance disparity (e.g., NDCG, Hit Rate) between sensitive groups, ensuring no group is systematically disadvantaged. This aligns with fairness goals like equalized odds. Since groups are defined by sensitive attributes, minimizing $\Delta F$ ensures equitable treatment, reflecting fairness for these attributes. While operationalized as prediction stability, the objective of balancing outcomes is consistent with classical fairness notions.
>
> ---
>
> ### **Question**: Is it possible to explicitly write  ....
>
> ### Answer
> We appreciate the reviewer’s question on explicitly writing out the objective function. Below, we present the full formulation and reference the equations in the submission.
>
> ### Objective Function
>
> The optimization objective is to find the thresholds  $\(\hat\lambda_{G_1}, \hat\lambda_{G_2} \)$ ( Line 215 ) such that:
>
> $$
> \hat\lambda_{G_1}, \hat\lambda_{G_2} = \sup ( \{ \lambda_{G_1}, \lambda_{G_2} \in [0, 1] \mid R^+(\lambda_G, \delta) \leq \alpha, \; \Delta F^+(\lambda_{G_1}, \lambda_{G_2}, \hat{\delta}) \leq \eta \} ),
> $$
>
> where:
> 1. $\(R^+(\lambda_G, \delta)\)$ is the upper bound of the risk metric (refer to the Risk Metric equation below).
> 2. $\(\Delta F^+(\lambda_{G_1}, \lambda_{G_2}, \hat{\delta})\)$ is the upper bound of fairness disparity (refer to the Fairness Metric equation below).
>
> ### Risk Metric
> The group-wise risk metric (Line  247) ensures that the true item is not excluded from the prediction set with high probability:
> $$
> R^+(\lambda_G, \delta) = \sup \left[ \hat{R}(\lambda_G) : \text{BinomCDF}(n\hat{R}(\lambda_G), n, \alpha) \leq \delta \right].
> $$
>
> This equation  guarantees that the group-level risk, $\(\hat{R}(\lambda_G)\),$ is statistically bounded with respect to the confidence parameter $\(\delta\)$.
>
>
> ### Fairness Metric
> The fairness disparity metric  (Line 260) is controlled using the following upper bound:
>
> $$
> \Delta F^+(\lambda_{G_1}, \lambda_{G_2}, \hat{\delta}) = \Delta F(\lambda_{G_1}, \lambda_{G_2}) + \sqrt{\frac{2 \sigma_F^2 \log \left(\frac{2}{\hat{\delta}}\right) + \frac{2}{3} \log \left(\frac{2}{\hat{\delta}}\right)}{n_1 + n_2}},
> $$
>
> where $\(\Delta F(\lambda_{G_1}, \lambda_{G_2})\)$ represents the fairness disparity between the two groups.
>
>
>
> ### Risk Definition
> The risk metric $\(R(\lambda_G)\) $ (line 251) is defined explicitly as:
>
> $$
> R(\lambda_G) = \frac{1}{|G|} \sum_{u \in G} L(i_{true}, \phi_{\lambda_G}(u)),
> $$
>
> where $\(L(i_{true}, \phi_{\lambda_G}(u))\)$ is the loss incurred if the true item $\(i_{true}\)$ is not in the prediction set $\(\phi_{\lambda_G}(u)\)$.
>
>
> ### Fairness Disparity Definition
> The fairness disparity metric $\(\Delta F(\lambda_{G_1}$, $\lambda_{G_2})\) $ (Line 161) is defined as:
>
> $$
> \Delta F(\lambda_{G_1}, \lambda_{G_2}) := \left| \frac{1}{|G_1|} \sum_{u \in G_1} M(\phi_{\lambda_{G_1}}(u)) - \frac{1}{|G_2|} \sum_{u \in G_2} M(\phi_{\lambda_{G_2}}(u)) \right|,
> $$
>
> where $\(M(\phi_{\lambda_G}(u))\) $ represents the metric of interest (e.g., Hit Rate or NDCG) for 2 user groups.
>
> ---
>
> ### Addressing Convexity and Practical Feasibility
>
> We sincerely appreciate the reviewer’s insightful comments regarding the convexity of the problem and the feasibility of finding an optimal solution. Below, we provide clarifications and additional details to address these concerns.
>
> Existence of a Solution
> - Assumption 2 (Line 665) guarantees the existence of a feasible $\lambda_{\text{min}}$ that satisfies both the risk and fairness constraints.
> - While $\lambda_{\text{min}}$ theoretically corresponds to including all items in the prediction set, such extremes are rarely required in practice. The algorithm converges well before reaching this boundary.
>
> Convexity and Monotonicity
> - Due to the monotonic nature of both the objective function and the constraints (risk and fairness metrics), the optimization can be efficiently reduced to an iterative grid-based search.
> - This structure allows the optimization to behave effectively as a convex problem within the bounded search space, ensuring that an optimal solution can always be found.
>
> Practical Feasibility
> - Our algorithm systematically adjusts $\lambda_{G_1}$ and $\lambda_{G_2}$ in small, incremental steps ($\Delta \lambda$), leveraging monotonicity to ensure efficient convergence.
> - Additionally, empirical results (Tables 1–4) consistently demonstrate that the algorithm is capable of deriving prediction sets that satisfy both constraints across all evaluated datasets.
>
> We are grateful to the reviewer for the opportunity to provide these explanations.
>
> [1] Li, Yunqi, et al. "User-oriented fairness in recommendation." WWW' 2021

---

> > ### Comment · Reviewer_NF9U · 2024-11-27
> >
> > Thanks for your response, and they address my concerns. I suggest the authors explicitly stating that the grouping is based on sensitive attributes in the problem formulation. I will raise my score to 6.

---

> > > ### Author Response · Authors · 2024-11-28
> > >
> > > We sincerely thank the reviewer for their positive feedback and for raising the score. We greatly appreciate the reviewer's suggestion to refine our problem to ensure the grouping assumption is clearly stated upfront. We will make the change included in the revised submission.
> > >
> > > Once again, we are grateful for your constructive feedback, which has helped us improve the clarity and rigor of our paper.
> > > Many Thanks.

---

### Official Review · Reviewer_U6ND · 2024-10-29

**Soundness:** 1
**Presentation:** 2
**Contribution:** 2
**Rating:** 6
**Confidence:** 5

**Summary:**

This paper addresses the critical issue of fairness in recommender systems, an area of increasing importance in machine learning and artificial intelligence. The authors propose an approach aimed at enhancing fairness in recommendations by introducing a framework that seeks to minimize the disparity in performance between advantaged and disadvantaged user groups. The paper provides a comprehensive overview of their methodology, including various modeling stages and experimental setups designed to evaluate the effectiveness of their proposed solution. The authors conduct experiments to prove the effectiveness of the proposed method.

**Strengths:**

1. Fairness is a crucial area of research in recommender systems, and the authors' focus on this topic is commendable.
2. The overall methodology presented in the paper is logical and coherent.

**Weaknesses:**

The authors demonstrate a lack of deep understanding and research in the field of fairness in recommender systems. Specific issues include the following:

1. The authors assert that existing methods "fail to construct a generalized fairness-based recommendation framework for different applications." However, the term "generalized fairness method" is not clearly defined. How does the proposed method differ from existing approaches? This statement lacks clarity and is not supported by references, theoretical backing, or experimental evidence.

2. The authors use interaction data to distinguish between advantaged and disadvantaged users, aiming to reduce the performance gap between these two groups. To my knowledge, this aligns closely with user-oriented fairness [1], with the only distinction being the constraint of the minimum prediction set in this paper. However, the authors do not compare their approach with any existing user-oriented fairness methods [2,3,4,5], nor do they analyze the advantages of their method over current approaches. It is essential to compare against existing methods in the experiments, and the authors could add constraints on the top-k values to introduce an average set size constraint for current methods.

3. While the paper aims to guarantee user fairness in recommendations, it only conducts experiments on user-oriented fairness without addressing attribute fairness, e.g., with gender or race as the sensitive attribute. Such experimental conclusions do not support the claim that the approach is generalized.

4. The figures in the paper are too small and unclear, making it difficult to interpret the results effectively.

[1] Li Y, Chen H, Fu Z, et al. User-oriented fairness in recommendation[C]//Proceedings of the web conference 2021. 2021: 624-632.\
[2] Han Z, Chen C, Zheng X, et al. In-processing User Constrained Dominant Sets for User-Oriented Fairness in Recommender Systems[C]//Proceedings of the 31st ACM International Conference on Multimedia. 2023: 6190-6201.\
[3] Liu Q, Feng X, Gu T, et al. FairCRS: Towards User-oriented Fairness in Conversational Recommendation Systems[C]//Proceedings of the 18th ACM Conference on Recommender Systems. 2024: 126-136.\
[4] Han Z, Chen C, Zheng X, et al. Hypergraph Convolutional Network for User-Oriented Fairness in Recommender Systems[C]//Proceedings of the 47th International ACM SIGIR Conference on Research and Development in Information Retrieval. 2024: 903-913.\
[5] Han Z, Chen C, Zheng X, et al. Intra-and Inter-group Optimal Transport for User-Oriented Fairness in Recommender Systems[C]//Proceedings of the AAAI Conference on Artificial Intelligence. 2024, 38(8): 8463-8471.

**Questions:**

See weakness.

---

> ### Author Response · Authors · 2024-11-24
>
> We sincerely thank the reviewer for their thoughtful and constructive feedback, which has been invaluable in improving our manuscript. Below, we address each concern.
>
> ---
>
> ## Question: The term "generalized fairness method" is not clearly defined. Additionally, how are the theoretical claims supported, and how do the experiments validate these claims?
>
> ### Answer:
> We acknowledge the reviewer’s concern that the term "generalized fairness method" was insufficiently defined. To clarify, we revised this in the manuscript (highlighted in red) and specified that no existing statistically guaranteed fairness model ensures fairness and accuracy across datasets and groupings.
>
> Theoretical Support:
> Existing fairness methods rely on heuristics, requiring dataset-specific tuning and assuming implicit data distributions. In contrast, our approach uses conformal prediction [1] and concentration inequalities [2] to provide statistical guarantees without distribution assumptions. We use Theorems 1 and 2 in Section 4 to establish upper bounds on risk and fairness, while Theorems 3 and 4 in Section 5 ensure we achieve confidence-controlled fairness and accuracy while minimizing prediction set sizes.
>
> Experimental Validation:
> We expanded our experiments to include diverse datasets and sensitive attributes: AmazonOffice (e-commerce, grouped by interactions), MovieLens (movies, grouped by gender), Last.fm (music, grouped by region, total interactions, and popular item consumption), and Book-Crossing (books, grouped by age). The results **(Section 6.3.2)** validate that the GUFR framework is a generalized fairness method as it consistently achieves fairness and risk guarantees across diverse contexts.
>
> Novelty:
> To the best of our knowledge, our work is the first to use conformal prediction to ensure fairness and accuracy in recommendation systems, addressing limitations of heuristic approaches, such as scalability and lack of statistical guarantees.
>
> ---
>
> ## Question: The paper does not compare the proposed method with existing user-oriented fairness methods.
>
> ### Answer:
> We thank the reviewer for raising this important concern. While GUFR is a post-processing method like user-oriented fairness [3], it introduces a dynamic prediction set mechanism with statistical guarantees for fairness and accuracy, distinguishing it from the heuristic-based reranking approach.
>
> Comparisons:
> To address this, we compared GUFR with additional in-processing [4] and post-processing [3] fairness baselines, fixing prediction set sizes for direct comparisons in Section 6.3.1. We further experimented with the time efficiency analysis in Section 6.3.3. The results depict GUFR outperforms these baselines in fairness and risk guarantees while being computationally more efficient.
>
> We are grateful to the reviewers for sharing the papers [5,6,7]. We acknowledge their importance in the domain of fairness in RS and have cited them accordingly in our updated document.
>
> ---
>
> ## Question: The paper claims to address generalized fairness but only evaluates user-oriented fairness, not attribute fairness (e.g., gender, race).
>
> ### Answer:
> On the reviewer’s suggestion, we have extended our evaluation to include attribute-based fairness using gender, age, and region as sensitive attributes (details are in Section 6.2).
>
> While race is an important attribute, we couldn’t find any well-known publicly available datasets, possibly due to privacy and ethical constraints. However, based on our results with other common sensitive attributes (region, gender, and age), we are confident that our system can effectively address biases related to race if suitable datasets become available.
>
> ---
>
> ## Question: Figures are too small and unclear, making results difficult to interpret.
>
> ### Answer:
> We have updated figures with larger font sizes for legends and labels to improve clarity and readability.
>
> ---
>
> We sincerely thank the reviewer for their thoughtful feedback, which has significantly improved the rigor and presentation of our work.
>
> ---
>
> ### References
>
> 1. Angelopoulos et al., A gentle introduction to conformal prediction and distribution-free uncertainty quantification.
> 2. Boucheron et al., Concentration Inequalities. Advanced Lectures on Machine Learning.
> 3. Li et al., User-oriented fairness in recommendation.
> 4. Rashidul Islam et al., Debiasing Career Recommendations with Neural Fair Collaborative Filtering.
> 5. Han et al., In-processing User Constrained Dominant Sets for User-Oriented Fairness in Recommender Systems.
> 6. Han et al., Hypergraph Convolutional Network for User-Oriented Fairness in Recommender Systems.
> 7. Han et al., Intra-and Inter-group Optimal Transport for User-Oriented Fairness in Recommender Systems.

---

> > ### Comment · Reviewer_U6ND · 2024-11-24
> > **Thanks for authors's rebuttal**
> >
> > Thank you for the detailed responses. While some of my concerns have been addressed, I still find the lack of comparisons with state-of-the-art (SOTA) methods concerning. The authors choose to compare with [1, 2] from 2021 instead of more recent SOTA methods such as [3, 4], making the experimental results unreliable.
> >
> > [1] Rashidul Islam et al., Debiasing Career Recommendations with Neural Fair Collaborative Filtering.
> > [2] Li et al., User-oriented fairness in recommendation.
> > [3] Han et al., In-processing User Constrained Dominant Sets for User-Oriented Fairness in Recommender Systems.
> > [4] Han et al., Hypergraph Convolutional Network for User-Oriented Fairness in Recommender Systems.

---

> ### Author Response · Authors · 2024-11-24
>
> We thank the reviewer for their thoughtful response. We are pleased that we were able to address some of their concerns and will attempt addressing their remaining concerns:
>
> a) We greatly appreciate the reviewer recommending the works [1][2][3][4] in their initial review and [1][2] again in this review. These papers are foundational to user-oriented fairness research, and we have cited them accordingly to acknowledge their significant contributions to advancing the field.
>
> However, we faced practical constraints in incorporating these methods:
>
> - For three papers [1][2][3], publicly available code was not accessible. Despite proactively contacting the authors twelve days ago to request implementation details, we have not yet received a response.
> - One paper [4] provided code, but its focus on conversational recommendation systems was not directly applicable to the objectives of our work.
>
> We respect the importance of these methods and remain committed to incorporating them into our future experiments once the necessary details become available.
>
> ---
>
> b) Given these constraints, we selected two accessible baselines to ensure a rigorous evaluation within the available timeframe:
>
> 1. A pre-processing method from 2021 [5], which provided both recency and relevance compared to the baseline [7] from 2020, cited in the shared works.
> 2. A post-processing method from 2021 [6], which was directly cited as a baseline in the shared works.
>
> While these baselines were chosen out of necessity rather than preference, they allowed us to robustly evaluate our approach in the context of established fairness methods. We remain committed to expanding our comparisons to include the reviewer-suggested methods in future work once implementation details are accessible.
>
> ---
>
> However, we emphasize that our work is the first to introduce a novel fairness framework leveraging conformal prediction and providing statistical guarantees for fairness and accuracy. This framework addresses critical gaps in heuristic-based methods, including those shared by the reviewer [1][2][3][4][6]. Our results highlight several strengths of our approach:
>
> - **Dataset-agnostic**: Effective across diverse datasets.
> - **Model-agnostic**: Applicable to various recommendation models.
> - **Group-agnostic and parameter-free**: Ensures fairness without requiring sensitive parameter tuning.
>
> By addressing these challenges with robust statistical guarantees, our framework builds upon the foundation established by the reviewer’s cited works while pushing the boundaries of fairness in recommendation systems.
>
> ---
>
> ## Conclusion
>
> We hope this response clarifies the rationale behind our baseline selection, proactive efforts to address the constraints, and the significant contributions of our work. We again thank the reviewer for their constructive feedback, which has been instrumental in refining our study, and we look forward to further advancing fairness research in collaboration with the broader community.
>
> ---
>
> **References**
>
> 1. Han Z, Chen C, Zheng X, et al. In-processing User-Constrained Dominant Sets for User-Oriented Fairness in Recommender Systems.
> 2. Han Z, Chen C, Zheng X, et al. Hypergraph Convolutional Network for User-Oriented Fairness in Recommender Systems.
> 3. Han Z, Chen C, Zheng X, et al. Intra- and Inter-Group Optimal Transport for User-Oriented Fairness in Recommender Systems.
> 4. Liu Q, Feng X, Gu T, et al. FairCRS: Towards User-Oriented Fairness in Conversational Recommendation Systems.
> 5. Rashidul Islam et al., Debiasing Career Recommendations with Neural Fair Collaborative Filtering.
> 6. Li et al., User-Oriented Fairness in Recommendation.
> 7. Wen et al., Distributionally-Robust Recommendations for Improving Worst-Case User Experience.

---

> > ### Comment · Reviewer_feQt · 2024-11-24
> > **Thanks for your prompt reply**
> >
> > Thank you for the authors' prompt response. I understand the challenges associated with the lack of publicly available code. The authors have adequately addressed my concerns, and I would like to raise my score from 3 to 6.

---

> > > ### Author Response · Authors · 2024-11-24
> > > **Thanks to the Reviewers**
> > >
> > > We sincerely thank the reviewer for their updated feedback and for raising the score. We appreciate the reviewers acknowledged the challenges we faced regarding code availability and are glad to have addressed your concerns adequately.
> > >
> > > Your constructive feedback has significantly strengthened our work, and we are committed to including the suggested comparisons in future research when implementation details become accessible.
> > >
> > > Thank you once again for your time and thoughtful engagement.
> > >
> > > Many Thanks

---

### Official Review · Reviewer_feQt · 2024-10-29

**Soundness:** 3
**Presentation:** 2
**Contribution:** 2
**Rating:** 5
**Confidence:** 4

**Summary:**

This paper addresses the important issue of fairness in recommender systems, proposing a novel framework aimed at improving fairness outcomes for users. The authors thoroughly analyze their proposed method, providing comprehensive proofs and assessments of its effectiveness. They conduct extensive hyperparameter experiments and make their source code publicly available, promoting transparency and reproducibility. Despite these contributions, the paper has several presentation and methodological shortcomings that require attention.

**Strengths:**

S1. The overall organization of the presentation is smooth, making it easy to read and understand the authors' intentions.\
S2. The authors provide a comprehensive proof and analysis of the proposed method.\
S3. The paper includes extensive hyperparameter experiments and openly shares the source code.\

**Weaknesses:**

W1. The paper exhibits numerous presentation detail issues, including:
1. There are noticeable typos, such as the one found in line 53: "However, ,".
2. Some equations lack punctuation following them.
3. The figures in the paper are of poor quality and too small to effectively convey information.

W2. Additionally, the experimental section presents significant issues:

1. Why do the authors differentiate users based on interactions rather than sensitive attributes, such as race?
2. The paper does not compare their approach with existing user-oriented fairness methods.
3. The authors only report the performance of the recommendation model combined with the GUFR method, without providing the original performance of the recommendation model. This omission makes it difficult to determine if GUFR actually enhances the original model's performance.

**Questions:**

My main concerns have already proposed in the weaknesses section, I would like to pose an additional question:\
Q1. Why is it necessary to guarantee the minimum average prediction set size? What significance does this have for recommender systems, and are there any related studies on this topic?

---

> ### Author Response · Authors · 2024-11-24
>
> We sincerely thank the reviewer for their thoughtful and constructive feedback, which has been invaluable in improving our manuscript. Below, we address each concern.
>
> ---
>
> ## Question: The paper exhibits presentation issues, including typos (e.g., line 53: "However, ,"), missing punctuation after equations, and figures that are too small and unclear.
>
> ### Answer:
> We thank the reviewer for highlighting these issues. We have corrected all typographical errors, including the one on line 53, and ensured that equations are properly punctuated. We also updated figures with larger font sizes for labels and legends for improved readability.
>
> ---
>
> ## Question: Why differentiate users based on interactions rather than sensitive attributes, such as race?
>
> ### Answer:
> We used interaction-based grouping [1] as it leverages readily available user engagement data, making it practical and adaptable across recommendation scenarios.
>
> To address the concern of the reviewer, we further conducted experiments on three more datasets ( Last.fm, MovieLens, and Book-Crossing) using three sensitive attributes: region, gender, and age. The results in **Section 6.3.1, Tables 2–4**, demonstrate that GUFR achieves fairness and risk guarantees across diverse contexts.
>
> While race is an important attribute, we couldn’t find any well-known publicly available datasets, possibly due to privacy and ethical constraints. However, based on our results with other common sensitive attributes (region, gender, and age), we are confident that our system can effectively address biases related to race if suitable datasets become available.
>
>
> ---
>
> ## Question: The paper does not compare the proposed method with existing user-oriented fairness methods.
>
> ### Answer:
> We thank the reviewer for highlighting this concern. To address this, we compared our framework on performance and time efficiency with several in-processing [2] and post-processing [1] fairness methods (e.g., NFCF, MFCF, NeuMF-UFR, GMF-UFR). The results in **Section 6.3.1, Tables 1–4, and Section 6.3.3, Table 5**, highlight that GUFR outperforms these baselines in fairness and risk guarantees while being computationally more efficient.
>
> ---
>
> ## Question: The authors only report the performance of the recommendation model with GUFR, without providing the original performance of the model.
>
> ### Answer:
> We have updated our Tables 1–4 in Section 6.3.1 with experiments comparing models with and without GUFR. The results highlight that GUFR consistently enhances fairness and ensures risk guarantees, while baseline models often fail in either of the two aspects. We have also uploaded relevant files and datasets in the codebase.
>
> ---
>
> ## Question: Why is it necessary to guarantee the minimum average prediction set size?
>
> ### Answer:
> We thank the reviewer for raising this very important point. Minimum set size guarantees are essential for user retention and the long-term sustainability of recommendation systems. Specifically, smaller sets ensure concise, personalized recommendations that meet user expectations, enhancing their experience. Additionally, from the platform perspective, it optimizes resources by reducing computational overhead and latency, thereby ensuring its sustainability.
>
> ---
>
> We appreciate your thoughtful feedback, which has significantly improved the rigor and clarity of our work. Thank you for your time and consideration.
>
> ---
>
> ### References:
>
> 1. Li et al. (2021, April). User-oriented fairness in recommendation. In Proceedings of the Web Conference 2021 (pp. 624-632).
> 2. Rashidul Islam et al., Debiasing Career Recommendations with Neural Fair Collaborative Filtering. (WWW '21).

---

> > ### Comment · Reviewer_feQt · 2024-11-24
> >
> > I believe that guaranteeing the minimum average prediction set size is not practical in real-world recommendation scenarios. In reality, recommendation systems typically maintain prediction sets of the same size for all users. The goal of better recommendation outcomes is not to minimize the average size of prediction sets but to maximize the likelihood of including items that users may interact with within a fixed prediction set size. Additionally, it is crucial to prioritize and rank items with higher probabilities of user interaction. Therefore, I suggest that the authors do not claim to guarantee the minimum average prediction set size as a challenge that should be solved. It is difficult to realize in real life. I will maintain my score.

---

> ### Author Response · Authors · 2024-11-26
>
> We sincerely thank the reviewer for their feedback and for highlighting practical considerations in real-world recommendation systems. Below, we address the concerns regarding the practicality of minimizing the average prediction set size:
>
>
>
> ### 1. The role of Prediction Set Size Minimization:
>
> Our work does not advocate for minimizing the prediction set size as a standalone objective but rather as part of a balance between user fairness and utility. Our approach does not compete with traditional fixed-size top-k recommendations but complements them. By guaranteeing a minimum average prediction set size, we aim to control the uncertainty in recommendations and ensure that all users receive meaningful and fair predictions. This approach complements ranking-based metrics like NDCG, which prioritize and rank items by relevance. While this paper initially didn't focus on real-world implementation of the minimum set size guarantee, the theoretical foundation it provides is directly extendable. One recent work by Kweon et al. [1] explores Top-Personalized-K Recommendation, aligning closely with our vision.
>
> ---
> ### 2. Relevance to Real-World Systems:
>
> In real-world, where recommendation systems typically use some arbitrary fixed size k  for all users, determined heuristically or based on trial and error. This may or may not represent the best choice , leading to a) Increased cognitive load for users, who may receive overly long lists of recommendations. b) Resource inefficiencies for platforms, particularly when unnecessary items are recommended. c) Fixed-size sets can also exacerbate performance and fairness issues by not accounting for individual user needs or disparities in group fairness. Our approach can solve the following concerns in following ways:
>
> a) Dynamic Determination of Optimal k:
>
> Given a validation set, our framework identifies the minimum prediction set size for each user that satisfies predefined performance and fairness criteria with statistical guarantees (e.g., 95% confidence).  To generalize across users, we can empirically employ an appropriate aggregation method (for exp. mean) to compute global k.  This global k, obtained with the theoretical guarantees, can then be applied to unseen users, ensuring that fairness and performance guarantees hold across the system.
>
> For exp : Consider an e-commerce platform like Amazon. Instead of heuristically fixing the prediction set size (e.g., k=10), our framework determined k (e.g., k=7) that balances fairness and accuracy for all users. By applying this k system-wide, the platform ensures that recommendations are concise, personalized, and resource-efficient without sacrificing fairness or performance ensuring increased user satisfaction, retention and platform’s  sustainability.
>
>
>
> b) Model Optimization and Fine-Tuning:
>
> The calculated average k can serve as a benchmark for selecting the best-performing model among candidates.  Alternatively, it can be used iteratively to fine-tune the underlying base recommendation model, enabling the system to achieve the desired levels of accuracy and fairness across users.
>
> For example: A video streaming platform like Netflix can use this approach to dynamically optimize k for new users, ensuring personalized recommendations that align with both user experience and platform resource constraints. Additionally, the feedback from k values can inform iterative improvements to their recommendation algorithms.
>
>
> ---
>
>
> ### 3. Revision in the paper:
>
> We deeply appreciate the reviewer's point about the practicality of minimizing the average prediction set size. In response, we have revised the manuscript to show its applicability (Appendix A.5) as a tool for enhancing fairness and reducing uncertainty, compatible with fixed-size prediction sets and ranking-based metrics.
>
> Reference:
>
> [1] Kweon, Wonbin, et al. "Top-Personalized-K Recommendation." Proceedings of the ACM on Web Conference 2024. 2024.

---

### Official Review · Reviewer_p37S · 2024-11-04

**Soundness:** 2
**Presentation:** 2
**Contribution:** 2
**Rating:** 3
**Confidence:** 2

**Summary:**

The authors propose an algorithm to determine the ranking length for personalized recommendation while ensuring risk-control and user-oriented fairness.
The proposed method is based on the framework of Risk-Controlling Prediction Sets (RCPS),
which allows for the straightforward analysis of statistical guarantees as provided in Sections 4 and 5.
The authors also design a greedy algorithm to efficiently optimize score thresholds based on their theoretical analysis.
However, some of the notations/definitions in the current manuscript are confusing, and the theoretical claims are thus not convincing.
The empirical evaluation also lacks baseline methods,
and I believe the reported results do not directly demonstrate the effectiveness of the proposed method.

**Strengths:**

1. The authors propose the concept of RCPS and FCPS for recommender systems.
2. The reproducible code is available.

**Weaknesses:**

### (W1) The problem formulation is somehow unrealistic.
The authors define the expected risk (Eq. (3)) based on the 0-1 loss for each user in Eq. (1),
which is similar to the hit rate measure.
To my understanding, the 0-1 loss serves as a measure of user dissatisfaction, and so,
minimizing the expectation of the 0-1 loss allows us to guarantee each user's satisfaction and achieve risk-control.
Nevertheless, they define the user fairness metric based on generalized recommendation measures for each user, which also serves as user satisfaction, rather than the hit rate measure.
It is quite counterintuitive for me because the user merit functions in the risk and fairness metrics are inconsistent.
In particular, since NDCG takes values between 0 and 1 and **not a function for a set** but an ordered set, the ranking position of a relevant item is essential in evaluating user satisfaction.
Also, using NDCG as the user loss is not trivial based on the currently provided theoretical guarantee;
the assumption of Bernoulli distributed losses in Theorem 1 is no longer valid.

### (W2) The empirical evaluation is insufficient.
In Section 6, the authors compare the different base models with the proposed method.
However, there is no comparison between the models with and without the proposed method,
and the current evaluation is not sufficient to directly show the effectiveness of the proposed method.
It would be helpful to set some baseline methods.

**Questions:**

### (Q1) On the definition of 0-1 user loss for multiple relevant items.
Please clarify the random variables to take expectation in Eq. (3).
In recommender systems, we often observe multiple relevant items for each user.
So, the current notation of $i_{true}$ is quite confusing.
How can the 0-1 loss be defined for multiple relevant items for a single user?

### (Q2) On the monotonicity of recommendation metrics.
In line 177, the authors compute NDCG for the set output of $\phi$.
However, NDCG is a ranking measure and cannot be computed for unordered sets; the hit rate measure is ok, on the other hand.
Also note that NDCG includes a normalization factor (i.e., ideal DCG) that may increase with the ranking length.
This implies that the monotonicity of $\Delta F$ assumed in line 681 is not trivial.
Authors can use DCG instead; it is additive and monotonic w.r.t. $\lambda$.
Still, I think that the monotonicity of $\Delta F$ is quite questionable;
even if $M(\phi_\lambda(u))$ is monotonic w.r.t. $\lambda$ for all $u$,
$\Delta F$ is generally not monotonic because of the absolute difference.
Any clarifications on this point would be helpful.

---

> ### Author Response · Authors · 2024-11-24
>
> We sincerely thank the reviewer for their thoughtful and constructive feedback. Below, we address the main concerns raised.
>
>
>
> ---
>
>
>
> ## Question: There seems to be inconsistency between the use of 0-1 loss... and the assumption of Bernoulli-distributed losses in Theorem 1...
>
>
>
> ### Answer:
>
> We thank the reviewer for raising this critical point. We utilize risk as a simple 0-1 loss to guarantee that each user group receives a minimally acceptable recommendation. This is critical in recommender systems as this measure ensures no user group is entirely dissatisfied. In contrast, the goal of fairness is to provide equitable outcomes across user groups (e.g., advantaged vs. disadvantaged users). We agree with reviewer that simply using hit rate diff. could simplify framework and ensure consistency however NDCG diff. is essential to capture ranking fairness, ensuring the quality of recommendations is consistent across groups. Therefore, we used performance disparity (in terms of Hit Rate Diff. and NDCG Diff.) as the fairness metric. Our choice of metrics is grounded in the existing literature on fairness in user-sided recommender systems [1]
>
>
> We wish to explain that Theorem 1 is used exclusively for risk guarantees associated with the 0-1 loss. While Theorem 1 ensures the true item is included in the prediction set, providing coverage and reliability for all user groups, as the reviewer pointed out, it is not valid for metrics like NDCG. Therefore for NDCG loss, we rely on Theorem 2, which employs Bernstein inequalities. These inequalities provide concentration bounds for metrics with non-binary outcomes and bounded variances like NDCG, unlike Bernoulli's assumption, which is restricted to binary loss.
>
>
>
> ---
>
>
>
> ## Question: The empirical evaluation lacks comparisons between models with and without the proposed method....
>
>
>
> ### Answer:
>
> We appreciate this important point, as it helped refine the empirical aspect of our work. To address this concern, we conducted additional experiments to compare the effectiveness of the proposed GUFR framework. Specifically:
>
>
>
> 1. Base Models Comparison: We compared base models with and without GUFR, demonstrating that GUFR enhances fairness and ensures risk guarantees while maintaining comparable performance.
>
> 2. Baseline Fairness Comparison: We evaluated GUFR against in-processing and post-processing fairness baselines. Results show GUFR consistently outperforms these baselines, which often fail to meet one or more fairness or risk thresholds.
>
> 3. Computational Efficiency: We analyzed computational efficiency, where GUFR demonstrated faster runtime than fairness baselines, validating its time-efficiency.
>
>
>
> The updated results **(Tables 1–5)** demonstrate GUFR’s effectiveness in ensuring fairness and reliability while being computationally efficient. All relevant code and data have been made available for reproducibility.
>
>
>
> ---
>
>
>
> ## Question: How is the 0-1 loss defined when users have multiple relevant items?
>
>
>
> ### Answer:
>
> Our framework currently adopts the Leave-One-Out (LOO) methodology, a standard in recommendation systems research [3][4], where each user has one true item in the test set. This setting simplifies evaluation and aligns with practical applications like music streaming or news recommendations.
>
>
>
> While our framework focuses on single-item scenarios,  we acknowledge extending it to handle multiple relevant items will further enhance the framework’s applicability to diverse contexts.
>
>
>
> ---
>
>
>
> ## Question: NDCG needs ordered set .... Additionally, monotonicity of NDCG is not trivial..
>
>
>
> ### Answer:
>
> We appreciate the reviewer for pointing this out. In our implementation, user-item interaction scores are sorted before computing NDCG, ensuring consistency with its definition as a ranking measure.
>
>
>
> Under LOO setting, NDCG is non-decreasing with the size of the prediction set. Specifically, as $\lambda$ decreases, more items are added in the prediction set, which increases the probability of selecting the relevant items and thereby improving NDCG. This satisfies the monotonicity condition required for the fairness metric ($\Delta$F). While alternative metrics like DCG could also be considered, NDCG aligns better with our framework's ranking quality and fairness goals.
>
>
>
> ---
>
>
>
> We sincerely appreciate the reviewer’s thoughtful feedback and hope our responses address the concerns raised. Thank you again for the opportunity to improve our work.
>
>
>
> ---
>
>
>
> ### References
>
>
>
> 1. Li, Yunqi, et al. User-oriented fairness in recommendation.
>
> 2. Islam, et al. Debiasing Career Recommendations with Neural Fair Collaborative Filtering.
>
> 3. Xu et al. (2023). Toward Equivalent Transformation of User Preferences in Cross Domain Recommendation
>
> 4. Han et al. (2023). In-processing User Constrained Dominant Sets for User-Oriented Fairness in Recommender Systems

---

> > ### Comment · Reviewer_p37S · 2024-11-25
> >
> > Thank authors for your detailed responses.
> > I appreciate authors' responses that indeed clarify some of my concerns. However, I will maintain my original score.
> > I would like to share my comments on authors' responses.
> >
> > First, regarding (W1), what I intended to express is that I do not understand the reasoning behind using a user loss/risk metric that differs from the ranking metric. These two metrics clearly measure the same concept, and the current setup feels arbitrary and unrealistic.
> > As for the theoretical analysis with non-Bernoulli losses, I agree with authors that it could be addressed using other probabilistic/concentration inequalities.
> >
> > The most important point, however, is that the adoption of Leave-One-Out (LOO) for experimental evaluation and the theoretical analysis should be clearly stated in the discussion. Making such assumptions implies that the proposed theory and method hold only in scenarios where there is exactly one positive item per user. This diverges significantly from the settings of many real-world applications, severely narrowing the applicability of the method. Providing theoretical guarantees under such unrealistic assumptions is, in my opinion, a minor contribution.
> >
> > As an aside, authors state, "While alternative metrics like DCG could also be considered, NDCG aligns better with our framework's ranking quality and fairness goals."
> > However, under the LOO assumption, nDCG is equivalent to DCG because the ideal DCG is 1 for all $K=1,\dots$.
> > I strongly recommend that authors first state the LOO assumption clearly and include the monotonicity of nDCG under LOO as a lemma.

---

> > > ### Author Response · Authors · 2024-11-26
> > >
> > > We thank the reviewer for their response and are glad we were able to alleviate few of their concerns. We appreciate their thoughtful and constructive feedback and appreciate the opportunity to address the concerns raised. Below, we attempt to answer the concerns:
> > >
> > >
> > > ### **Question** . ...the current setup feels arbitrary and unrealistic....
> > >
> > > We appreciate the reviewer’s observations regarding the use of 0-1 loss for risk control and NDCG for fairness and we concede we might have missed to answer this in previous reply with enough clarity. This choice we made is intentional to address two distinct yet complementary goals a) Risk Control: Ensuring minimum reliability by guaranteeing that at least one relevant item is included in the prediction set for all users. b) Fairness: Measuring disparities in ranking quality across user groups using NDCG to ensure equity.
> > >
> > > These metrics are designed to work together, with 0-1 loss providing reliability and NDCG capturing ranking equity between the user groups. Combining them allows for a more nuanced framework. While aligning the metrics (e.g., the same metric for both) could simplify the setup, it would overlook this balance, potentially compromising the framework's ability to address both goals effectively.
> > >
> > > ---
> > >
> > > ### **Question** ...making such assumptions implies that the proposed theory...  settings of many real-world applications..
> > >
> > > We sincerely appreciate the reviewer’s concern about the LOO assumption and its impact on the generality of our framework. To clarify, we employ the LOO assumption solely in our experimental evaluation, as we mentioned in the experimental setup,  aligning with established practices in recommendation research [1][2]. However, we respectfully disagree with the reviewer that this assumption limits the theoretical contributions of our work. Our framework's guarantees are based on the statistical properties of the risk and fairness metrics, developed using theories of concentration inequalities and conformal prediction methods. These guarantees are probabilistic and are not dependent on the LOO setup, ensuring broader applicability.
> > >
> > > While the LOO setup might be simplistic, it has direct relevance in many real-world recommender system contexts, such as:
> > >
> > > - Sequential recommendations (e.g., predicting the next relevant item like a song or product),
> > > - Cold-start scenarios (e.g., recommending the first item to a user),
> > > - Online learning and adversarial robustness tasks,
> > > - Anomaly detection in recommendation systems,
> > > - Exploration-exploitation trade-off problem in recommendations etc.
> > >
> > > These examples highlight the practicality of the LOO setup for multiple scenarios where our framework can be readily applied.
> > >
> > > We acknowledge the reviewer’s suggestion to explore a more generalized framework for multi-item settings. Thanks to the flexible theoretical foundation of our approach, extending it to scenarios involving multiple relevant items can be achieved with minor modifications. This adaptability underscores the broader potential of our framework and demonstrates its capacity to handle diverse recommendation scenarios, which we aim to explore in greater detail in future work.
> > >
> > > ---
> > >
> > >
> > >
> > > ### 3. **Question** nDCG is equivalent to DCG...
> > >
> > > We thank the reviewer for highlighting the equivalence of NDCG and DCG under the LOO setting, given that the iDCG is always 1. While this is true in our current evaluation setup, we chose NDCG for its recognition in recommendation research and its ability to generalize scenarios with multiple relevant items. This choice ensures consistency with established practices while preparing the framework for broader applications beyond the LOO setting.
> > >
> > > ---
> > >
> > > ### 4.**Question** ..Providing theoretical guarantees.. in my opinion, a minor contribution..
> > >
> > > We appreciate the opportunity to clarify the theoretical contributions of our work. Most existing fairness approaches rely on heuristics or dataset-specific adjustments, which can limit their generalizability. In contrast, our framework introduces statistical guarantees for fairness and risk metrics through conformal prediction and concentration inequalities. These guarantees are designed to be model- and dataset-agnostic, enabling applicability across a wide range of recommendation settings. By providing a formal foundation for fairness-aware recommendation systems, our work aims to address existing gaps in the literature and contribute to ongoing efforts in developing reliable and equitable recommendation systems.
> > >
> > > ---
> > >
> > > We again thank the reviewer for the thoughtful feedback, which has greatly helped us explain our contributions.
> > >
> > > ### References
> > > [1] Xu et al. (2023). Toward Equivalent Transformation of User Preferences in Cross Domain Recommendation
> > >
> > > [2] Han et al. (2023). In-processing User Constrained Dominant Sets for User-Oriented Fairness in Recommender Systems

---

> > > > ### Comment · Reviewer_p37S · 2024-11-27
> > > >
> > > > Thank authors for your responses.
> > > >
> > > > Choosing ranking metrics such as nDCG is equivalent to making assumptions about the user/examination model. In my opinion, it is unnatural to define the minimum reliability for users independently of that user model.
> > > > The current setup lacks a clear rationale apart from simplifying the theory and is not based on practical benefits.
> > > >
> > > > The claim, "These guarantees are probabilistic and are not dependent on the LOO setup, ensuring broader applicability," is false. This is because the monotonicity w.r.t. $\lambda$ used throughout the proof (which is the same as the monotonicity w.r.t. ranking length) is only valid for some ranking metrics (e.g., nDCG) under the LOO setup.
> > > >
> > > > I agree that there can be cases where the LOO setup is appropriate for certain applications. However, the methodologies used in the experiments and the recommender methods being compared are those prevalent in the field of collaborative filtering, and in the task of collaborative filtering, the LOO setup is neither common nor practical.
> > > >
> > > > Regarding the claim, "extending it to scenarios involving multiple relevant items can be achieved with minor modifications," I cannot agree with this. The proposed method assumes the monotonicity of a ranking metric with respect to ranking length for any rankings. Most such ranking metrics are unidimensional ones, such as recall, which do not imply precision. Therefore, I believe the proposed framework lacks generality.
> > > >
> > > >
> > > > I believe that theoretical contributions in this field are very important. However, providing guarantees by significantly altering the setup from a practical one to an unrealistic one diminishes the significance of that contribution. If technical difficulties require changing the setup to something unrealistic, I think it is important to acknowledge that change and engage in careful discussion about it.

---

> > > > > ### Author Response · Authors · 2024-11-28
> > > > >
> > > > > We sincerely thank the reviewer for their valuable time, insightful comments, and dedicated effort in reviewing our work.
> > > > >
> > > > > Our main contribution is to develop a risk-controlled framework with user-pre-defined confidence (i.e., say, 90% probability) via a calibration step, which can work on top of any recommendation model. Specifically, given a user model that generates relevance scores, we use these scores to compute standard metrics such as NDCG, and based on these metrics, we can optimize model parameter (e.g., λ) that align with stakeholder-defined thresholds to ensure both performance and fairness. Through rigorous theoretical analysis, we provide probabilistic guarantees on achieving the desired reliability, while our experimental results substantiate our claims that we achieve the desired reliability in metrics by creating predictions sets based on the calculated λ.
> > > > >
> > > > > We would like to clarify that the Leave-One-Out (LOO) setup mentioned by the reviewer is not inherently tied to our proposed framework. Instead, it is a data-splitting technique commonly employed in fairness literature [1][2][3], which is specified in line 820 of Appendix A.3.2. The framework has already formulated the problem setup by clarifying the only one true item in Section 3, and the theory is to demonstrate the validity of the proposed framework, which has no relationship with the concept of LOO, as the LOO is served as a practical technique choice in experiments for data splitting.
> > > > >
> > > > > As our framework is a post-processing method, it can work on top of any recommendation model, including collaborative filtering methods. We embark on this fundamental research from the only one true item setting with rigorous theoretical guarantee and extensive experiments, and we believe it does make a big leap forward to ensuring guaranteed reliability and fairness in recommender systems. We are committed to addressing the scenarios involving multiple relevant items by relaxing monotonicity conditions in the next work.
> > > > >
> > > > > Once again, we are grateful for the reviewer’s constructive feedback. We will follow the reviewer’s suggestions to add a discussion regarding the limit of the current setting and the vision of future extension, improving the clarity and readability of the manuscript.
> > > > >
> > > > > ### References
> > > > > [1] He, Xiangnan, et al. "Neural collaborative filtering." Proceedings of the 26th international conference on world wide web. 2017.
> > > > > [2] Han, Z., Chen, C., Zheng, X., Liu, W., Wang, J., Cheng, W., & Li, Y. (2023, October). In-processing User Constrained Dominant Sets for User-Oriented Fairness in Recommender Systems. In Proceedings of the 31st ACM International Conference on Multimedia (pp. 6190-6201).
> > > > > [3] Chen, X., Zhang, Y., Tsang, I. W., Pan, Y., & Su, J. (2023). Toward equivalent transformation of user preferences in cross domain recommendation. ACM Transactions on Information Systems, 41(1), 1-31.

---

### Note · Authors · 2025-01-24

I have read and agree with the venue's withdrawal policy on behalf of myself and my co-authors.